# Assessing the severity of laparotomy and partial hepatectomy in male rats—A multimodal approach

**Leonie Zieglowski**[1], **Anna Maria Kümmecke**[1], **Lisa Ernst**[1], **Rupert Palme**[2], **Ralf Weiskirchen**[3], **Steven R. Talbot**[4], **René H. Tolba**[1]*

**1** Institute for Laboratory Animal Science & Experimental Surgery, Faculty of Medicine, RWTH Aachen University, Aachen, Germany, **2** Department of Biomedical Sciences, University of Veterinary Medicine, Vienna, Austria, **3** Institute of Molecular Pathobiochemistry, Faculty of Medicine, Experimental Gene Therapy and Clinical Chemistry (IFMPEGKC), RWTH Aachen University, Aachen, Germany, **4** Institute for Laboratory Animal Science, Hannover Medical School, Hannover, Germany

* rtolba@ukaachen.de

**Data Availability Statement:** All addition data refered to in the manuscript, are available from the online data repository Zenodo: https://doi.org/10.

## Abstract

This study assessed the postoperative severity after three different visceral surgical interventions in rats by using objective parameters pertaining to various disciplines. The objective was to evaluate whether the degree of severity increases with the invasiveness of the intervention and whether this is in accordance with the EU Directive 2010/63. 136 adult male WistarHan rats were assigned to three groups: Sham-laparotomy (Sham) [7 days post-surgical survival time]; 50% partial hepatectomy (PH); 70% PH [PH groups with 1, 3, or 7 days post-surgical survival times]. Post-surgical severity assessment was performed via several multimodal assessment tools: I) model-specific score sheet focusing on body weight, general condition, spontaneous behavior, and the animals' willingness to move as well as on wound healing; II) Open Field tests evaluating the total distance and velocity an animal moved within 10 minutes and its rearing behavior during the test; III) telemetric data analyzing heart rate and blood pressure; and IV) analysis of blood (AST, ALT, and hemogram) and fecal samples (fecal corticosterone metabolites). Significant differences among the experimental groups and models were observed. We demonstrated that the Open Field test can detect significant changes in severity levels. Sham-laparotomy and removal of 50% of the liver mass were associated with comparable severity (mild–moderate); the severity parameters returned to baseline levels within seven days. Removal of 70% of the liver tissue seemed to be associated with a moderate severity grade and entailed a longer recovery period (>7 days) for complete regeneration. We recommend the use of Open Field tests as part of multimodal objective severity assessment.

## Introduction

The European Union (EU) Directive 2010/63/EU mandates adherence to the principle of the "Three Rs" (reduction, refinement, replacement) and establishment of severity assessments for

5281/zenodo.3842977 or https://zenodo.org/record/3842977.

**Funding:** RT funded by Deutsche Forschungsgemeinschaft, TO 542/5-1 (https://www.dfg.de/gefoerderte_projekte/programme_und_projekte/listen/projektdetails/index.jsp?id=321137804). The funders had no role in study design, data collection and analysis, decision to publish, or preparation of the manuscript.

**Competing interests:** The authors declare no competing interests.

animal research [1]. All scientific research projects are currently required to categorize the expected and the retrospective actual degree of severity of their protocols and classify these as mild, moderate, or severe (as defined in Annex VIII of the Directive). However, standardization of animal species- and trial-specific parameters and development of objective evaluation criteria are key requirements for this purpose.

The key elements of a valid severity assessment pertain to three main domains: 1.) "Biochemistry and Biomarker", 2.) "Physiology and Clinic", and 3.) "Behavior" [2]. During an experiment, animals are liable to be influenced by a wide variety of factors. Therefore, it is necessary to identify the stressful states that can be recorded, assessed, and classified [3]. In this study, we evaluated the use of existing assessment parameters for developing a more objective, model-specific approach for the assessment of severity levels in animal research. As part of the German research group FOR 2591 "Severity Assessment in animal-based Research," the experiments presented here intend to assess severity after partial liver resection in male rats. This surgical model is widely used in oncology, hepatology, and endocrinology research [4]. Liver resection in rats was first described by Higgins and Anderson in 1931 and has since been used in many different ways [5]. The most commonly used rat models of liver resection entail removal of 65–72% of the total liver parenchyma in male Wistar rats. The surgical technique described by Higgins and Anderson is still considered state-of-the-art after all these years [4]. However, Annex VIII of the EU Directive 2010/63/EU classifies the severity of general surgical procedures such as laparotomy, organ resection, and organ transplantation as "moderate." The postoperative burden on the animals invariably depends on several factors including the skills of the surgeon, the extent of liver resection, and the peri- and postoperative management and care of the animals. Therefore, the use of validated parameters to determine the degree of severity in an individual animal and its impact on animal behavior is a key imperative. For this purpose, assessment criteria applied in this study represent a selection of the available methods from the respective underlying disciplines, i.e., "biochemistry and biomarker" (blood and organ profiles, fecal corticosterone metabolites), "physiology and clinic" (scoring, telemetry, body weight), and "behavior" (Open Field, OF). This multimodal approach employs a composite scoring system to facilitate a comprehensive assessment of the current severity of the animals.

As a further method, behavioral testing via the OF test was used to assess postoperative severity in this surgical model involving the use of living animals as described elsewhere [6]. Here, the focus was set on to the evaluation of spontaneous locomotor behavior and voluntary willingness to move as indices of surgery-related severity in the animal.

## Material and methods

### Animals and ethics statement

All experiments were conducted in accordance with the German animal welfare law (Tierschutzgesetz, TSchG) and the EU Directive (2010/63/EU) [1]. The study protocol was approved by the Governmental Animal Care and Use Committee (Reference No.: 84–02.04.2017.A304; Landesamt für Natur, Umwelt und Verbraucherschutz Recklinghausen, Nordrhein-Westfalen, Germany; dated: 14.02.2018). The study protocol also complied with the Guide for the Care and Use of Laboratory Animals [7]. All staff members were trained in the rat-specific behavior and care and the corresponding handling methods, in advance. For this purpose, either special trainings according to FELASA guidelines were absolved or individual permits for the respective tasks were given by the supervisory authority.

A total of 136 male WistarHan rats (Janvier S.A.S., Saint-Berthevin Cedex, France) [delivery body weight (BW): 150–175 g; age range: 6–8 weeks] were used as this was identified to be the

most commonly used model in liver resection research [4]. The animals were kept under SPF-conditions according to FELASA recommendations [8]. All animals were group-housed in filter-top cages (Type 2000, Tecniplast, Buguggiate, Italy) and a controlled environment (12-h/12-h light-dark cycle; temperature: 22°C ± 2°C; relative humidity: 30%–70%). For cage enrichment, low-dust wood granulate was used as bedding (Rettenmeier Holding AG, Wilburgstetten, Germany) in addition to nesting material (Nestlet, 14010, Plexx B.V., Elst, The Netherlands). To reduce stress, play tunnels were used for handling, cage changes, or transfers to test setups (tunnel Ø 155 × 75 mm, #3084014, Zoonlab GmbH, Castrop-Rauxel, Germany). All rats were provided *ad libitum* access to standard diet (rat/mouse maintenance #V1534-300, 10 mm; ssniff Spezialdiäten GmbH, Soest, Germany) and sweetened (Ja! Süßstoff flüssig, Rio Mints & Sweeteners B.V., Utrecht, The Netherlands) drinking water. Liquid sweetener was used to mask the bitter taste of postoperatively administered analgesics (metamizole/dipyrone).

## Experimental setup

After seven days of acclimatization, the rats were randomly allocated to the laparotomy group (Sham, n = 10), 50% PH group (n = 63), or 70% PH group (n = 63) with different survival time-points (Table 1).

Rats were trained to the OF test thrice every alternate training day (D-20, D-18, and D-16). On D-15 (surgery I), a telemetric transmitter (hereafter referred to as transmitter implantation, TI; Data Sciences International, Minnesota, USA; HD-S11) was surgically implanted subcutaneously in the left flank in 12 animals of each 7-day survival group. All animals in the Sham group were implanted with a dummy device, as these animals were part of a pilot study [6] performed before studying PH. Transmitter implantation was followed by a 12-day recovery phase with no further intervention. To ensure comparability, this test-free recovery period was standardized for all animals, even if they were not telemetrically monitored. Thereafter, retraining was performed on D-2 and -1 in all animals, followed by Sham, 50% PH, or 70% PH (surgery II), on D0. OF tests were performed in the housing room and during the initial 3 h after the beginning of the light phase on POD1, POD3, POD4, and POD7 (depending on survival time) (Fig 1). Testing was performed only by female researchers to minimize any confounding influence on the results of the behavioral test [9].

Body weight was measured before individual training, as well as before and after each surgery; in addition, the body weight was measured once daily during postoperative scoring. Postoperative scoring was conducted three times a day from POD1 to POD3 and once a day in the morning from POD4 to POD7. The schedule of this experiment is identical to that used in the pilot study [6]. On euthanasia day (POD1, POD3, or POD7) rats were reopened (surgery III)

**Table 1. Overview of sample size, survival time, and implanted telemetric devices in various animal groups.**

| Study group | Survival time | Telemetric device | Rats (n) |
|---|---|---|---|
| Sham-laparotomy (Sham) | POD7 | n = 10 | 10 |
| 50% partial hepatectomy (50% PH) | POD1 | - | 21 |
| | POD3 | - | 21 |
| | POD7 | n = 12 | 21 |
| 70% partial hepatectomy (70% PH) | POD1 | - | 21 |
| | POD3 | - | 21 |
| | POD7 | n = 12 | 21 |

POD, postoperative day.

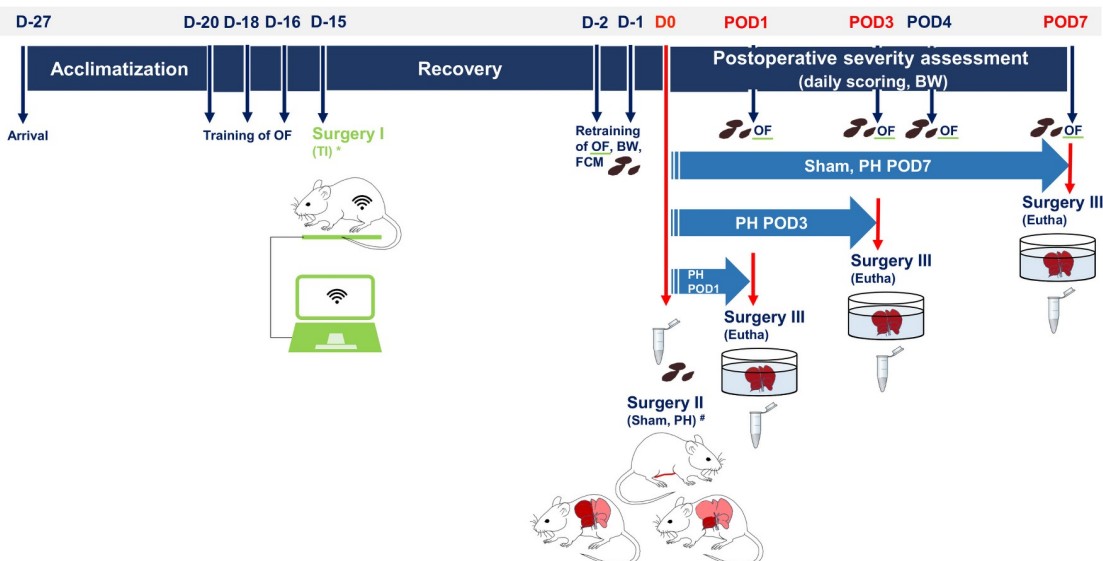

**Fig 1. Schematic illustration of the study timeline.** D: study day; POD: postoperative day; OF: Open Field test; TI: transmitter implantation (*:only in Sham and n = 12 animals of each POD7 survival group of PH); BW, body weight control; FCM, fecal sampling for analyses of fecal corticosterone metabolites; Sham, sham-laparotomy; PH, partial hepatectomy ([#]: 50% or 70% removal); eutha, the time point of euthanasia (Sham: n = 10 on POD7; PH groups: n = 21 each on POD1, POD3, or POD7); Eppendorf tube, blood sampling.

under general anesthesia [induction: 5 vol% isoflurane (Abbott GmbH, Wiesbaden, Germany) + 5 L $O_2$/min; maintenance: 2 vol% isoflurane + 2 L $O_2$/min] and analgesia (metamizole; Novaminsulfon-ratiopharm[®] 1 g/2 mL; Ratiopharm GmbH, Ulm, Germany; 100 mg/kg, s.c., single dose) and were euthanized by final blood withdrawal from the inferior *vena cava*. Organs were removed for subsequent examinations and processed accordingly.

## Open Field tests

For the OF test, rats were placed in the middle of the field (L 72 × W 72 × H 40 cm; water-resistant plastic with dark underground) and then video recorded from above for 10 min (Media Recorder 4, NOLDUS, Wageningen, The Netherlands; camera: Camera GigE monochrome, 1/ 1″; lens: Lens Std CS mount, 4.5–12.5 mm 1/2″, Basler AG, Ahrensburg, Germany) without further adaptation time. Analyses were performed using the NOLDUS EthoVision XT 14 software (NOLDUS, Wageningen, The Netherlands) with a focus on velocity, distance, and unsupported rearing. After each test run, the OF surface was cleaned, disinfected with wet wipes (Incidin Perfekt 5%, #104206E; Ecolab Deutschland GmbH, Germany), and subsequently wiped off.

## Surgical procedures and treatment

All surgeries (I–III) were performed in the same timeframe in a separate operating room under aseptic conditions and general anesthesia [induction: 5 vol% isoflurane (Abbott GmbH) + 5 L $O_2$/min; maintenance: 2 vol% isoflurane + 2 L $O_2$/min] with analgesics (metamizole; Novaminsulfon-ratiopharm[®] 1 g/2 mL; Ratiopharm GmbH, Germany; 100 mg/kg, s.c., single dose) and antibiotics (cefuroxime, 16 mg/kg, s.c.). Postoperative analgesia was based on the recommendations of the German Society for Laboratory Animal Science (GV-SOLAS) as well

as the Initiative Veterinary Pain Therapy [10, 11]. Blood samples were obtained during surgeries from the vena sublingualis (TI) or inferior vena cava (PH and euthanasia).

**Surgery I (D-15).** Transmitters were implanted (TI) subcutaneously in the left flank, with its blood pressure catheter placed in the femoral artery. ECG electrodes were led subcutaneously to the pectoral region and sutured to the muscle tissue. Dummy transmitters were identical in construction but were not equipped with a valid measuring unit.

**Surgery II (D0).** Sham and PH were performed via laparotomy (midline incision from the xiphoid to cranial pelvic brim, approximately 5 cm). Subsequently, blood samples were obtained from the inferior vena cava (750 μL). Within the Sham groups, liver lobes were carefully touched with a moistened cotton swab. After about 20 minutes, the abdominal cavity was rinsed with warm saline and closed in the same way as in PH groups. Here, the individual liver lobes to be resected were removed, according to their percentage of organ weight (Fig 2 and Table 2).

For 50% PH these were: left lateral lobe (LLL), left part of the medial hepatic lobe (LML), and both caudate lobes (CL). For 70% PH, LLL and ML were resected. The focus was not to resect the exact amount of 50% or 70% of the liver mass, but rather to find approximate values that at the same time represent a significant difference between the resected groups, which was based on the most frequently used resection models and resected liver lobes, according to Zieglowski et al. [4]. The isolated liver lobes were tied with 4/0 silk (Resorba Medical GmbH, Nürnberg, Germany) at the hilus of the lobe, cut off with scissors, and the remaining stump cauterized with bipolar forceps (HF-Generator, MBC, Söring GmbH, Quickborn, Germany) to avoid further bleeding. Subsequently, the abdominal cavity was rinsed with sterile saline at body temperature and blood clots removed; in the absence of any further bleeding, the abdomen was closed using a two-layer continuous suture (5/0 Prolene, Johnson & Johnson Medical GmbH ETHICON, Norderstedt, Germany) for the muscle layer and an interrupted suture for the skin (4/0 Vicryl, Johnson & Johnson Medical GmbH ETHICON). During the recovery phase, the animals were placed in a pre-oxygenated and warmed intensive care unit (Vetario; Brinsea Products Ltd., North Somerset, UK) and administered a single subcutaneous injection of sterile saline (10 mL/kg). To prevent wound infections, antibiotic therapy (cefuroxime, 16 mg/kg, s.c., once daily until POD 3) was administered right before and within the first three days post surgery. Analgesia was administered subcutaneously (metamizole, Novaminsulfon-ratiopharm® 1 g/2 mL; 100 mg/kg, s.c., single dose) prior to each surgery. For post-surgical pain management, metamizole (metamizole, Novaminsulfon-ratiopharm® 1 g/2 mL; 400 mg/kg/day, oral) was administered until the late afternoon of the third day after surgery (POD 3), via sweetened drinking water. This should enable an assessment on the morning of the fourth post-surgical day (POD 4), without the influence of analgesic medication.

**Surgery III (POD1, POD3, or POD7).** After abdominal re-opening and evaluation of the situs, animals were euthanized by final blood withdrawal from the inferior *vena cava*. The organs were removed, weighed, examined, and processed for further analysis. Euthanasia was set as the endpoint of the study on POD1, POD3, or POD7, depending on the survival group.

### Scores for the degree of severity

For severity scoring, a modified version of the scoring system described by Morton et al. (1985) [12] was used daily to assess the general condition of rats (see detailed score sheet in S1 File). Due to the number of surgeries performed, the member size of the working group, and the additional limitation of relying on female-only staff with sufficient expertise, the personnel were not blinded to the animals' treatment. The scoring was carried out in the home cage first, without further interference from outside, hereafter each animal was examined in detail,

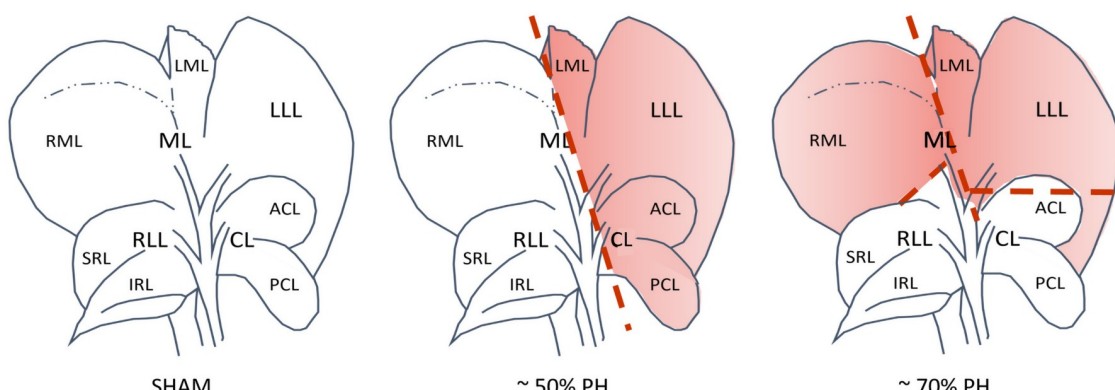

**Fig 2. Short name of individual liver lobes (visceral view).** From left to right: native liver (Sham), 50% PH, and 70% PH; resected liver lobes shown in red with resection line in the resection groups.

outside the cage. Here, the four major criteria were: 1) body weight; 2) overall state; 3) spontaneous locomotor behavior and readiness to walk; and 4) surgical procedure and wound healing. The spontaneous locomotor behavior was usually assessed first. To assess wound healing, several stages with subsequent proceedings were defined. If only the thread ends were gnawed off (wound edges adapted), a more intensive monitoring was performed. If the adaptation of wound edges was about to loosen, a wound clip was applied on the corresponding area to secure the suture, and was removed as soon as the wound looked resilient. If a suture dehiscence of single knots occured, the animal was re-anesthetized, the wound was cleaned, and re-sutured. In the case that the suture of the abdominal muscle layer ruptured (hernia; skin suture closed), the animal was re-anesthetized, skin sutures were removed and finally muscles layer and skin were re-sutured. Each criterion was scored on a scale of 1 to 20 [DS = 0 points (no distress), DS $\geq$ 5 points (mild distress), DS $\geq$ 10 points (moderate distress), and DS $\geq$ 20 points (severe distress)]. Rats with a morning score of $\geq$ 15 were not exposed to the OF on that day and were assessed more frequently (3 times per day). For a score of $\geq$ 15 points, a time limit of 24 hours was set as a humane endpoint. On the respective postoperative examination time-points, score points were summed up and were required to be $\leq$ 19 points to fall below the predefined humane endpoint of 20 score points. Further study-specific criteria to determine a humane endpoint of an individual animal were: body weight decrease $\geq$ 20% (in comparison to post-surgical body weight); cramps; paralysis; breathing difficulties; icterus; diarrhea (>48 h); animal feels cold; permanent crouching with closed eyes; repetitive suture

**Table 2. Liver lobes and their respective percentage in organ weight.**

| Liver lobe | Sub lobes | Total liver mass (%) |
|---|---|---|
| Left lateral lobe (LLL) | - | ~30 |
| Median lobes (ML) | left median lobe (LML) | ~38 |
| | right median lobe (RML) | |
| Right liver lobes (RLL) | inferior right lobe (IRL) | ~22 |
| | superior right lobe (SRL) | |
| Caudate lobes (CL) | anterior caudal lobe (ACL) | ~10 |
| | posterior caudal lobe (PCL) | |

(according to Zieglowski) [4].

dehiscence; severely infected wound; hemoperitoneum; severe tongue swelling with opened mouth. For statistical analysis, the individual scores on each POD were shown as boxplots.

## Measurement of serum and blood parameters

Serum levels of lactate dehydrogenase (LDH), aspartate aminotransferase (AST), and alanine aminotransferase (ALT) were determined as functional parameters of hepatocellular damage using a VITROS 350® Integrated System Analyzer (Ortho Clinical Diagnostics, NJ, USA) on surgery II and III (POD1, POD3, or POD7). For hematology, EDTA blood was analyzed and blood cell counts were determined using the Celltac α MEK-6450 K (Nihon Kohden Europe GmbH, Rosbach vor der Höhe, Germany); differential hemograms were determined and blood smears were microscopically evaluated after eosin methylene blue staining.

## Measurement of FCMs

Levels of corticosterone or its metabolites were determined in serum and fecal samples. Here, during OF or surgery weaned fecal samples were taken and analyzed using an enzyme immunoassay, as described elsewhere [13–15]. Due to the species-specific time delay between a stress-induced increase in plasma cortisol levels and respective corticosterone metabolites levels in feces (FCMs; peak approximately after 12 h), samples represented the impact of the intervention on the preceding day.

## Telemetric data acquisition

Data acquisition of heart rate and blood pressure were initiated by touching the animal with a magnet over the implanted transmitter, which leads to activation of the transmitter. Data were collected during the 10 minutes in the OF and the activity phase of the animals (overnight; 1h before the start of the dark phase until 1h after the end of the dark phase). To avoid anesthesia-related distortion of the measured parameters, the post-surgical overnight measurement was recorded from POD1 to POD2, first. This was followed by measurements from POD2 to POD3, from POD3 to POD4, and from POD6 to POD7. Therefore, the measured values are subsequently evaluated as data on the follow-up day (POD2, POD3, POD4, and POD7). Telemetric data were continuously collected during the measurement period and analyzed (Ponemah 6.4, Data Sciences International, USA). The parameters evaluated were heart rate (HR) and blood pressure (BP) and values of each measurement time (OF or overnight) were averaged with a logging rate of 1 minute.

## Statistical analysis

The sample size of animals in each group was determined using the open-source software G∗Power (version 3.1; Freeware, Heinrich Heine University of Düsseldorf, Düsseldorf, Germany; www.gpower.hhu.de). Before statistical analysis, the normality of distribution of all variables was assessed by Shapiro-Wilk, D'Agostino–Pearson, and Kolmogorov–Smirnov tests using the GraphPad Prism software (Windows version 7.04; GraphPad Software, San Diego, CA, USA). Depending on the outcomes of tests, non-parametric or parametric analyses were used. For non-parametric tests, Kruskal-Wallis and Dunn's post-hoc tests were chosen. Analysis of variance was performed using one-way or two-way ANOVA followed by post-hoc tests for multiple comparisons. Tukey's and Dunnett's tests were used for multi-group comparisons (e.g., comparison with Sham group) or comparison with baseline (Dunnett's). We used the Šidák correction for the comparisons of two groups (50% PH vs. 70% PH) at a particular time-point (PODs). Line graphs were prepared using the mean value ± standard deviation.

Post-hoc tests with p ≤ 0.05 were considered significant. Here, the following symbols represent the significances between the groups: "#" = Sham vs. 50% PH; "*" = Sham vs. 70% PH and "&" = 50% vs. 70% PH. Moreover, the quantity of symbols corresponds to the level of significance: # / */ & = p ≤ 0.05 up to ####/ ****/ &&&& = p ≤ 0.0001. Due to random variation in animals, test results were normalized to their baseline values, except for liver enzymes. Serum levels of ALT and AST are presented as units per liter. A Principal Component Analysis (PCA) was performed using the *factoextra* package [16] in the R software [17]. PCA calculation was limited to variables for which complete data were available. Therefore, only factor combinations that were available in all comparison groups at the same time-point were tested, scaled, and plotted.

Retraining values of OF tests (D-2 and D-1) were averaged; the mean values were set as baseline levels for behavioral analyses. Score points for the severity degree are shown as median with upper limits. Baseline values for blood and serum analysis were averaged out of samples obtained during the second surgery (Sham or PH). For FCM analysis, values of weaned feces from the second retraining were averaged and used as baselines. For the standardized calculation of FCMs/g feces in relation to liver weight (calculated from own data of body weight-liver weight-ratio), we used a value of 4.15%. Data were checked for normality and homoscedasticity using the Shapiro-Wilk and the Levene test. In case of violated model assumptions, the non-parametric Kruskal-Wallis test was preferred over ANOVA. Multiple group comparisons were performed using either Tukey's (ANOVA) or Dunn's test (Kruskal-Wallis). In the case of Dunn's test, the false-discovery rate for multiple testing was controlled with the Benjamini-Hochberg criterion. Data of re-sutured animals or animals with surgery-related complications were included in the analysis, as long as they did not reach the predefined humane endpoints.

## Results

### Determination of body weight and scores for degree of severity

After laparotomy or liver resection, body weight in all groups showed a decrease from POD1 until POD3 latest (70% PH), compared to post-surgical baseline values. While Sham animals gained weight after POD2 and reached baseline on POD3, 50% PH group exceeded the baseline level on POD4 and the median weight in the 70% PH group reached the initial post-surgical weight on POD5. Significant differences were observed between post-surgical weight changes in the Sham and PH groups at nearly every time-point (Fig 3). After reaching their lowest weight values, all groups showed a constant gain in weight until the sacrifice day.

On postoperative follow-up, the severity scores showed significant differences between the three groups (Fig 4). In the Sham and 50% PH group, all animals reached the intended end of the study; however, in the 70% PH group, two animals died due to major bleeding during PH and one animal had to be euthanized during post-surgical recovery phase due to stenosis (animal feels cold, esp. hind legs and paws; incipient paralysis). Three other animals died within the first three postoperative days (POD). Of these three animals, one was euthanized because it reached a humane endpoint (hemoperitoneum on POD1) and two died in the intervening nights of POD1 and POD3 (cause of death unknown). Further wound healing disorders occurred in the post-surgical recovery period: two skin sutures had to be fixed with additive wound clips (PH70%); one skin suture had to be re-sutured under short general anesthesia (SHAM group); six animals got a hernia of the muscle layer and had to be re-sutured under general anesthesia (PH70%). Overall, the scores of the Sham animals did not exceed the mild level (in median) at any time and the increase in severity scores only became noticeable on POD4 and POD5; however, the PH groups showed the peak mean value on POD2 (maximum

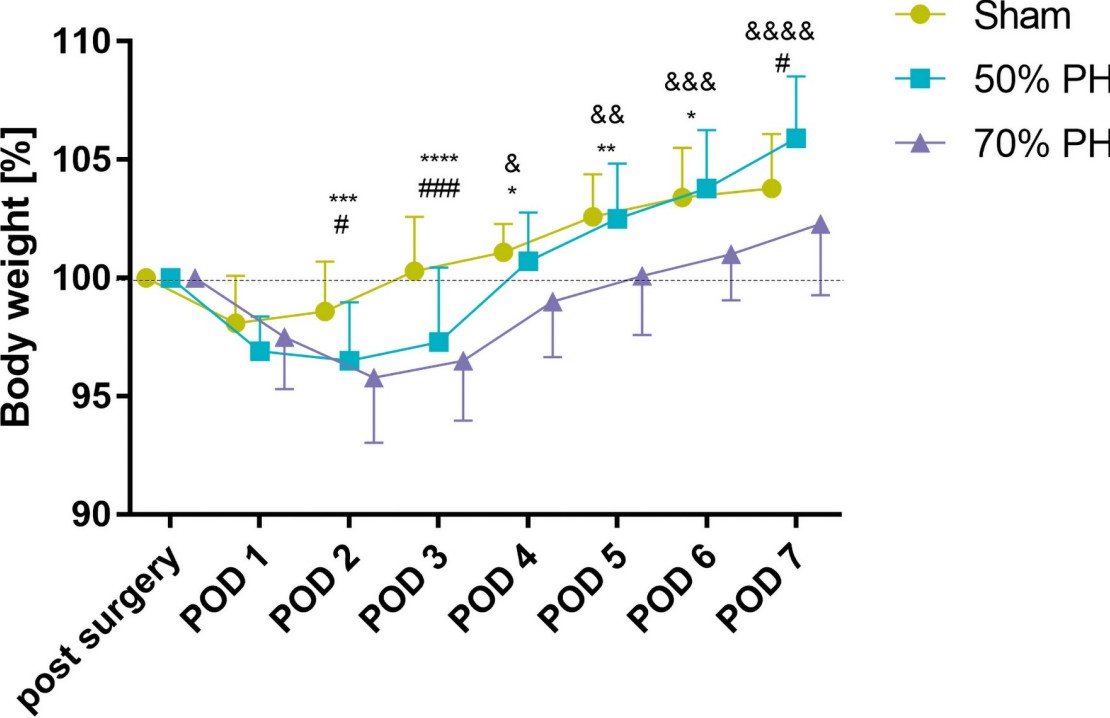

**Fig 3. Comparative body weight change (%) in the three study groups (Sham, 50% PH, 70% PH) during post-surgical phase.**
Animal numbers (Sham/ 50% PH / 70% PH): post-surgery (n = 10/63/59), POD 1 (n = 10/63/59), POD 2 (n = 10/42/39), POD 3 (n = 10/42/38), POD 4 (n = 10/21/18), POD 5 (n = 10/21/18), POD 6 (n = 10/21/18), POD 7 (n = 10/21/18); two-way ANOVA F (14,621) = 4.026, $p < 0.0001$; POD2: Sham vs. 50% PH (#) $p_{adj} = 0.0125$ and Sham vs. 70% PH (***) $p_{adj} = 0.0005$; POD3: Sham vs. 50% PH (###) $p_{adj} = 0.0002$ and Sham vs. 70% PH (****) $p_{adj} = < 0.0001$; POD4: Sham vs. 70% PH (*) $p_{adj} = 0.03$ and 50% PH vs. 70% PH (&) $p_{adj} = 0.0313$; POD5: Sham vs. 70% PH (**) $p_{adj} = 0.0072$ and 50% PH vs. 70% PH (&&) $p_{adj} = 0.0011$; POD6: Sham vs. 70% PH (*) $p_{adj} = 0.0105$ and 50% PH vs. 70% PH (&&&) $p_{adj} = 0.0001$; POD7: Sham vs. 50% PH (#) $p_{adj} = 0.0250$ and 50% PH vs. 70% PH (&&&&) $p_{adj} < 0.0001$.

value in both groups: 6 score points). On no other POD, a higher average score than 6 points was achieved.

### Open Field performance

To analyze the behavioral changes due to post-surgical severity, we investigated the OF performance with a focus on velocity, distance, and unsupported rearing behavior. The distance covered by the animals as well as the recorded velocity showed no significant differences between the Sham and PH groups. Similarly, no significant between-group differences were identified with respect to the frequency of unsupported rearing at any time-point (Fig 5). As shown in the graphs, all the assessed parameters showed almost identical postoperative progression. Only the median values in the 70% PH group did not reach the initial values up to POD7.

### Fecal corticosterone metabolites (FCMs)

In all groups, the peak level of FCMs [expressed as a percentage in relation to the second retraining (D-1)] was observed on POD1. On POD7, FCM levels had decreased to baseline

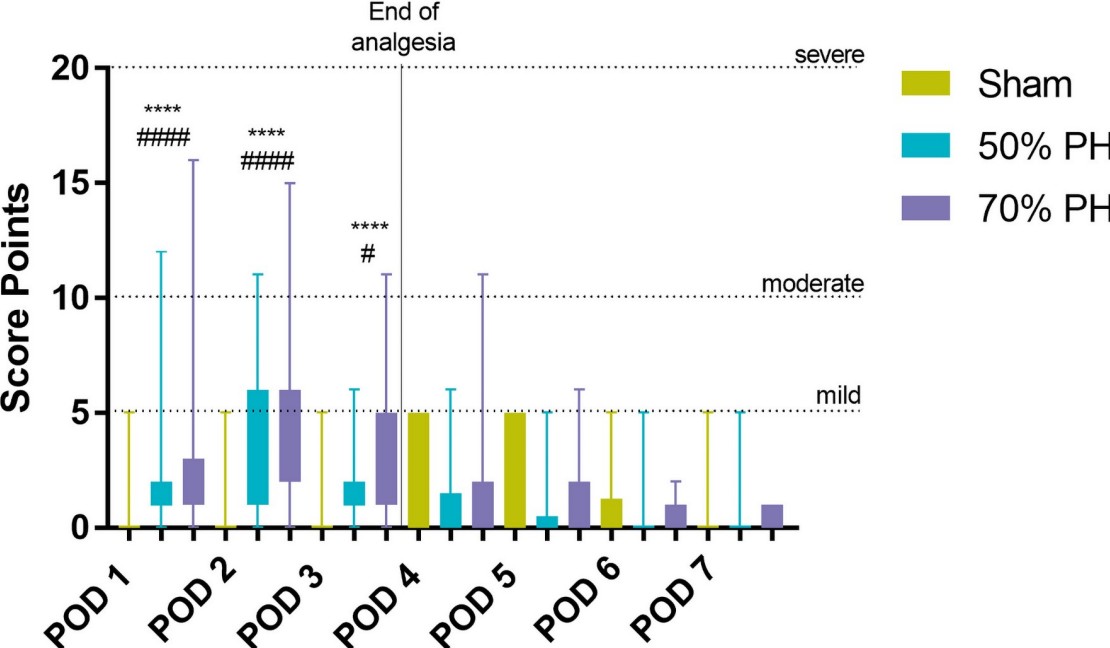

**Fig 4. Boxplots of severity scores.** (Body weight, general condition, wound healing, and spontaneous locomotor behavior) after Sham or PH with upper limits with a gradual allocation of severity (mild: $\geq$ 5 points, moderate: $\geq$ 10 points, and severe: $\geq$ 20 points); animal numbers (Sham/ 50% PH / 70% PH): POD 1 (n = 10/63/60), POD 2 (n = 10/42/39), POD 3 (n = 10/42/38), POD 4 (n = 10/21/18), POD 5 (n = 10/21/18), POD 6 (n = 10/21/18), POD 7 (n = 10/21/18); Kruskal–Wallis-test ($\chi^2$ = 353.3, $p$ < 0.0001, degrees of freedom (df) = 21); POD1: Sham vs. 50% PH (####) and Sham vs. 70% PH (****) $p_{adj}$ < 0.0001; POD2: Sham vs. 50% PH (####) and Sham vs. 70% PH (****) $p_{adj}$ < 0.0001; POD3: Sham vs. 50% PH (#) $p_{adj}$ = 0.0253 and Sham vs. 70% PH (****) $p_{adj}$ < 0.0001.

level in Sham animals and were lower than baseline levels in both resection groups (Fig 6A). After standardization of FCM levels, depending on the individual body weight-liver weight-ratio (BW-LW-ratio) at POD1, the Sham group exhibited the lowest level (Fig 6B). For the calculation of the ratio, a value of 4.15% liver weight (calculated from own data) in relation to total body weight, was used.

We performed a Principal Component Analysis (PCA) to investigate whether the above-mentioned parameters (body weight, severity score, OF distance, velocity, unsupported rearing, and FCMs) were influenced by variance. The three OF parameters were found to be the main contributors to the total variance representation in the first dimension. Therefore, these are relevant in all three study groups (Fig 7).

## Measurement of serum parameters and blood cell counts

Serum levels of ALT and AST (Fig 8), as well as LDH (data available in the online repository), were analyzed. All groups showed a nearly identical trend of change in liver enzyme levels. In all groups, all parameters showed a significant increase on POD1 and returned to baseline level latest by POD7.

Metamizole is known to cause agranulocytosis in patients [18], which we could not confirm in our animals. To identify and verify metamizole-dependent changes, which could have an impact on severity grades, the blood cell counts were assessed. On the evaluation of blood cell

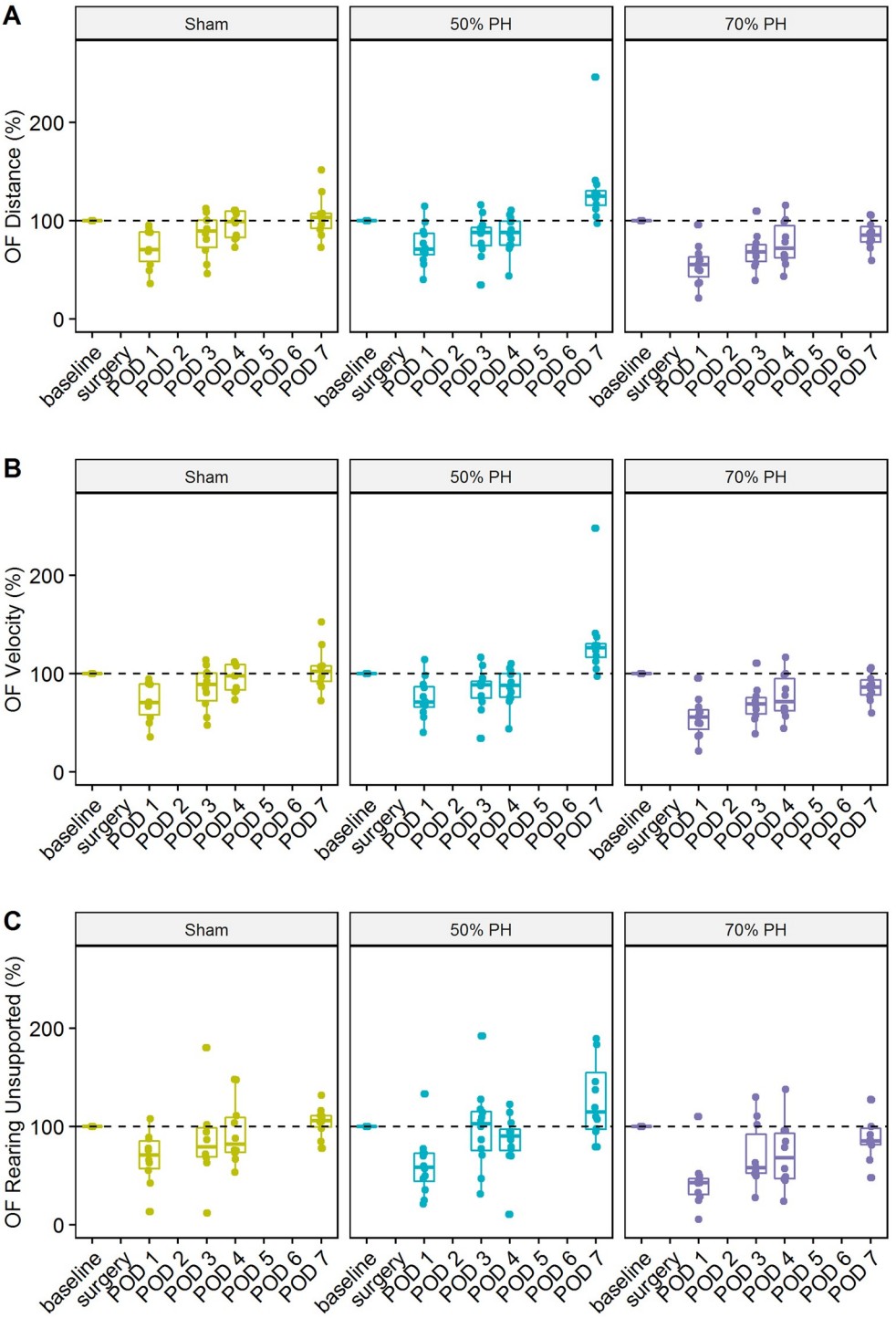

**Fig 5. Comparison of the course of Open Field modalities in various groups (Sham; 50% PH; 70% PH).** Boxplots of covered distance (A), velocity (B), and counts of unsupported rearing (C) during 10 min of test duration; no significant differences.

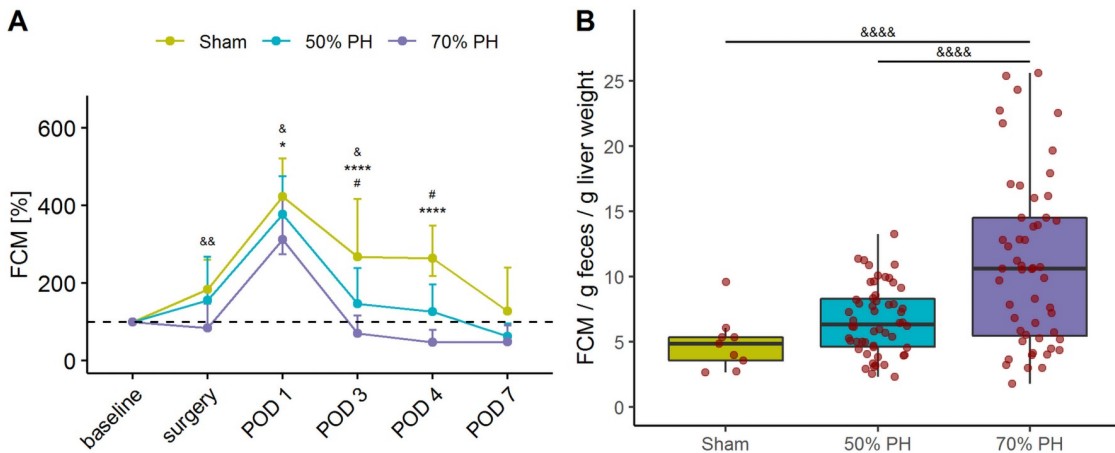

**Fig 6. Comparison of fecal corticosterone metabolites (FCMs) in samples weaned during Open Field or during anesthetic induction in various groups.** (A) Percentage changes in FCM levels in the three groups over time; sample sizes (Sham/ 50% PH / 70% PH) on each time point: baseline (n = 10/61/62), surgery (n = 10/54/57), POD 1 (n = 9/54/55), POD 3 (n = 10/40/36), POD 4 (n = 10/20/18), POD 7 (n = 10/19/18); F(10,528) = 2.062, $p < 0.0001$; surgery: 50% PH vs. 70% PH (&&) $p_{adj}$ = 0.0031; POD1: Sham vs. 70% PH (*) $p_{adj}$ = 0.0377 and 50% PH vs. 70% PH (&) $p_{adj}$ = 0.0299; POD3: Sham vs. 50% PH (#) $p_{adj}$ = 0.0169; Sham vs. 70% PH (****) $p_{adj}$ < 0.0001 and 50% PH vs. 70% PH (&) $p_{adj}$ = 0.0273; POD4: Sham vs. 50% PH (#) $p_{adj}$ = 0.0137 and Sham vs. 70% PH (****) $p_{adj}$ < 0.0001. (B) Standardization of FCM levels (μg/g feces) in relation to liver weight (calculated by BW-LW-ratio) at POD1; sample sizes: Sham n = 9, 50% PH n = 54, 70% PH n = 55. For liver weight, a value of 4.15% of total body weight was calculated (own data). Data were checked for normality and homoscedasticity using the Shapiro–Wilk and the Levene test. Owing to the observed variance between groups ($p = 0.007$), the non-parametric Kruskal-Wallis test was used ($\chi^2$ = 20.9, df = 2, $p < 0.0001$). The observed between-group differences were analyzed using the Dunn's post-hoc test and the false-discovery rate for multiple tests was controlled using the Benjamini-Hochberg criterion. Significant differences were found between the 50% and the 70% PH group (Z = -3.727, $p_{adj}$ = 0.0006) as well as between the 70% PH and the Sham group (Z = 3.541, $p_{ad}$ = 0.0006).

counts, significant between-group differences were observed only with respect to platelet and leucocyte counts. For leucocytes, mild differences were detected between 50% PH and 70% PH groups on POD1 ($p < 0.05$) and POD3 ($p < 0.01$). Platelet counts differed significantly between 50% PH and 70% PH on POD1 ($p < 0.01$) and highly significantly on POD3 ($p < 0.0001$); on POD7, platelet counts differed significantly between the Sham and 70% PH groups ($p < 0.05$). Leucocyte and platelet counts showed a similar trend in the different groups (data available in the online repository).

We also performed PCA to demonstrate the relevance of group comparisons of serum parameters (i.e., ALT, AST, and LDH) as well as blood cell counts (red blood cells, RBC; white blood cells, WBC; lymphocytes). The results showed that the analyzed blood counts at the different euthanasia time-points (POD1, POD3, and POD7) (Fig 9A) were distinguishable both in terms of their level and group membership (50% PH vs. 70% PH) (Fig 9B). Further, the liver enzymes (ALT, AST, and LDH) and differential segmented granulocytes were the four main contributors to the observed variance in both PH groups (Fig 9C).

### Telemetric data acquisition

Telemetry data were collected and evaluated during the postoperative phase, both overnight and during the 10-minute observation in the OF. With PCA, we examined the contributions of each variable to the two shown dimensions. Here, we compared the information contribution of telemetric data during OF testing, with the parameters "distance" and "unsupported rearing" of the OF test itself and the latest body weight. In comparison, telemetric data and body weight were found less relevant than OF parameters (Fig 10B and 10D). Furthermore, a general difference between the two treatments (50% PH and 70% PH) could be detected, both

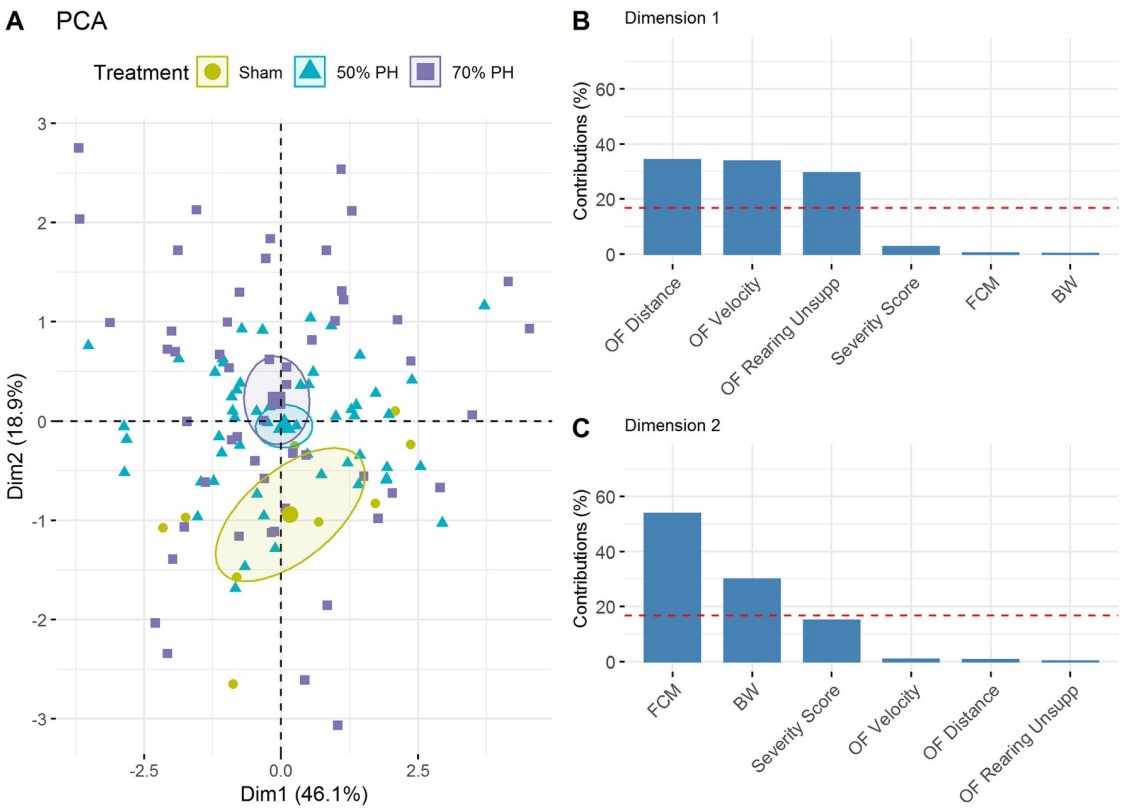

**Fig 7. Principal Component Analysis (PCA) of the Sham, 50% and 70% PH groups.** (A) Projection of individual treatment groups into the two-dimensional PCA factor space; group centroids are characterized by the 95% confidence ellipses. (B) Variance contributions of factors in percentage in the first and (C) second dimensions. BW, body weight; FCMs, fecal corticosterone metabolites (µg/g); OF, Open Field; OF RearingUnsupp, rearing unsupported; the dashed line shows the cut-off for the uniform variance distribution.

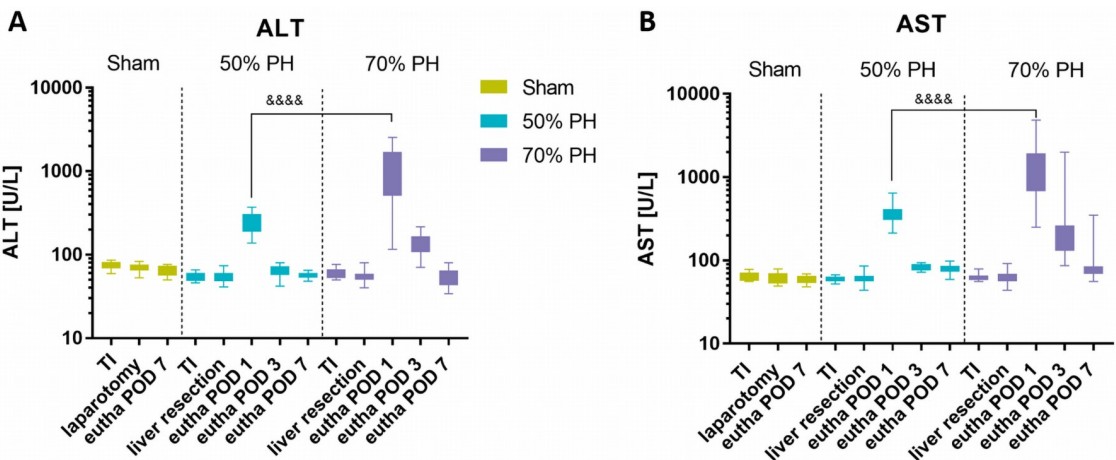

**Fig 8. Serum measurements.** Boxplots of serum (A) alanine aminotransferase (ALT) and (B) aspartate aminotransferase (AST) levels (U/L) showing group-individual values in group comparisons (Sham, 50% PH, 70% PH); Sham: TI (n = 9), laparotomy (n = 10), eutha POD 7 (n = 10); 50% PH: TI (n = 12), liver resection (n = 63), eutha POD 1 (n = 21), eutha POD 3 (n = 21), eutha POD 7 (n = 21); 70% PH: TI (n = 12), liver resection (n = 61), eutha POD 1 (n = 18), eutha POD 3 (n = 20), eutha POD 7 (n = 18); ALT: one-way ANOVA F(12,277) = 54.71, $p < 0.0001$; POD1: 50% PH vs. 70% PH (&&&&) $p_{adj} < 0.0001$. AST: one-way ANOVA F(12,276) = 26.05, $p < 0.0001$; POD1: 50% PH vs. 70% PH (&&&&) $p_{adj} < 0.0001$.

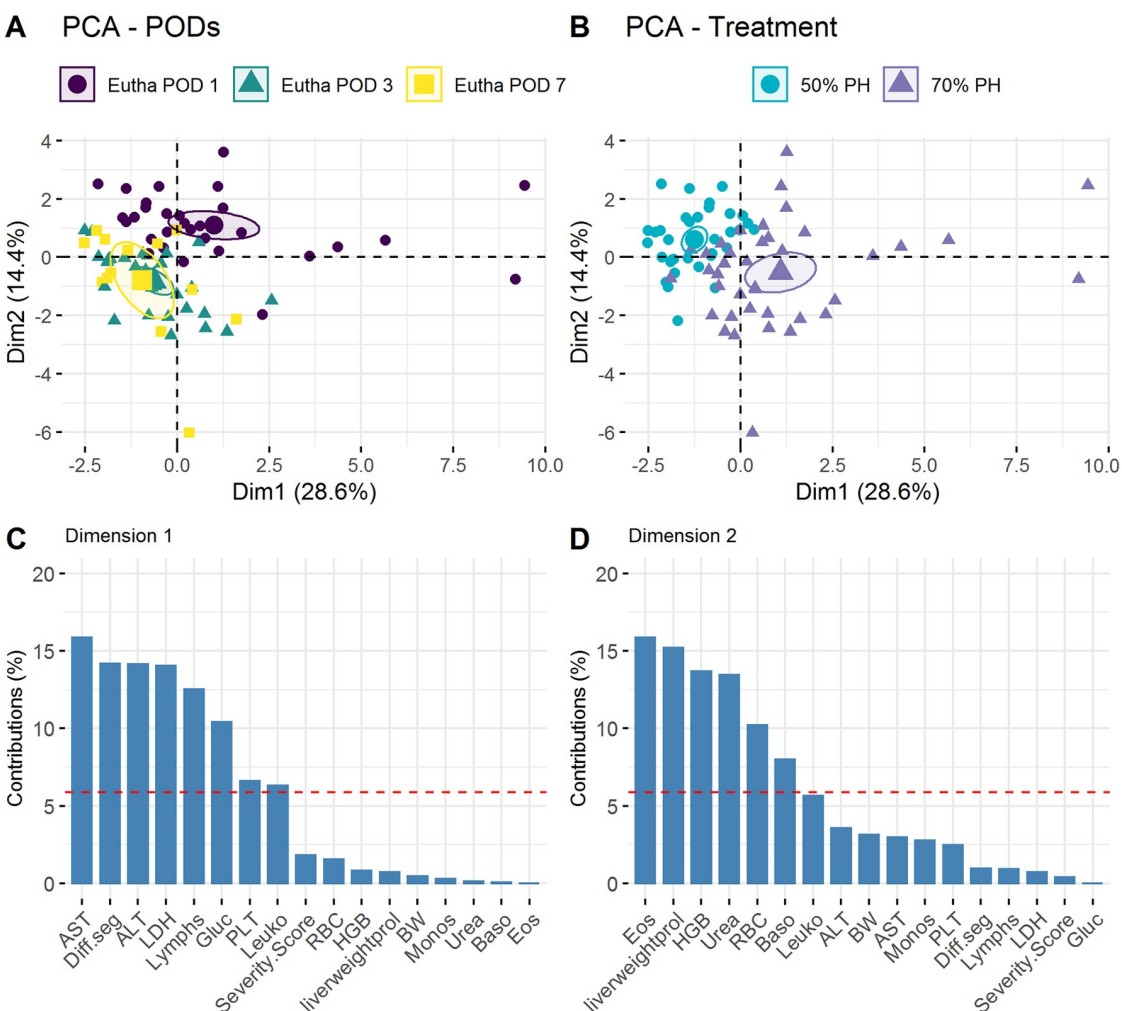

**Fig 9. Principal Component Analysis of blood and serum parameters of 50% and 70% PH groups.** (A) Projection of individual euthanasia time-points (POD1, POD3. or POD7) into the two-dimensional PCA factor space; group centroids are characterized by the 95% confidence ellipses. (B) Projection of the two different treatment groups (50% PH and 70% PH) into the two-dimensional PCA factor space; group centroids are characterized by the 95% confidence ellipses. (C) Variance contributions of factors in % in the first and (D) second dimensions. AST, alanine aminotransferase; ALT, aspartate aminotransferase; Baso, basophils; BW, body weight; Eos, eosinophils; Gluc, glucose; HGB, hemoglobin; LDH, lactate dehydrogenase; Leuko, leukocyte count; PLT, platelets; RBC, red blood cell counts; the dashed line shows the cut-off for the uniform variance distribution.

in daily comparisons of the measured heart rates of the animals in each group (Fig 10A and 10C) as well as in direct group comparisons (Fig 11A). Both groups showed changes along a horizontal axis over time. While the 50% PH group even exceeded the basal value on POD7, the 70% PH group did not return to their initial values by the end of the experiment.

In the overnight measurements (HR ON; Fig 11B), the lowest mean heart rate (HR) in 50% PH was found on POD2, whereas in the 70% PH group, the lowest mean HR was observed on POD3. The median HR in both groups did not return to baseline by the end of the study. The lowest HR in the OF (HR OF; Fig 11C) in both PH groups was observed on POD1, which returned to and exceeded the respective initial values by POD7.

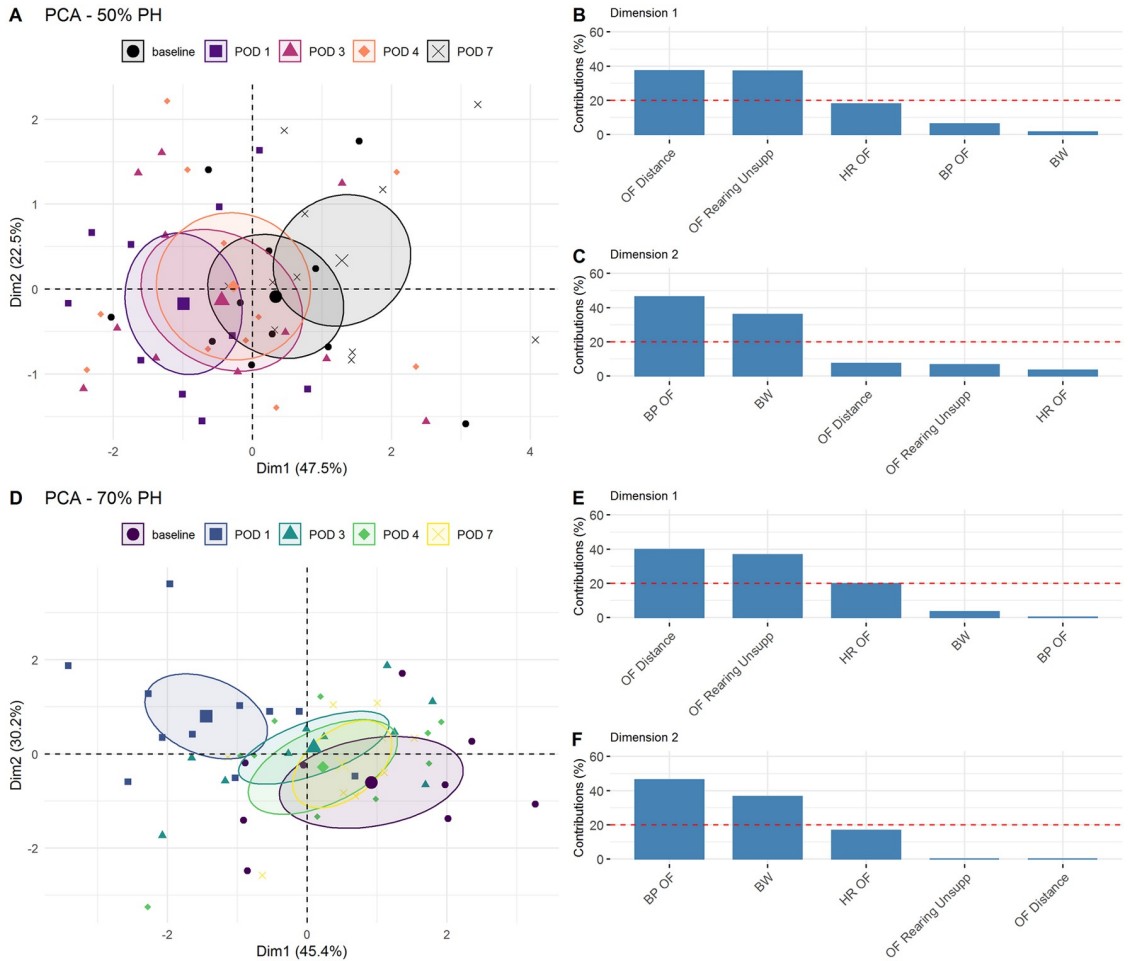

**Fig 10. Principal Component Analysis of telemetric data including body weight, telemetric data, and OF parameters in the 50% PH and 70% PH groups.** (A) Projection of different test time-points (baseline, POD1, POD3, POD4, and POD7) of 50% PH group into the two-dimensional PCA factor space; group centroids are characterized by the 95% confidence ellipses. Variance contributions of factors (A) in % in the first (B) and second (C) dimensions. (D) Projection of different test time-points (baseline, POD1, POD3, POD4, and POD7) of 70% PH group into the two-dimensional PCA factor space; group centroids are characterized by the 95% confidence ellipses. Variance contributions of factors (C) in % in the first (E) and second (F) dimensions. BP, blood pressure; BW, body weight; HR, heart rate; OF, Open Field; the dashed line shows the cut-off for the uniform variance distribution.

## Discussion

Annexure VIII of the EU Directive 2010/63 classifies intervention-related degrees of severity in laboratory animals as "mild," "moderate," and "severe" [1]. Surgical interventions are defined as more than punctiform incision of the skin. Therefore, surgical interventions are generally classified as "moderate," as far as they qualify the following criteria: "surgery under general anesthesia and appropriate analgesia associated with post-surgical pain, suffering or impairment of general condition". However, this definition does not provide further gradual differentiation of surgical interventions with regard to their potential impact on the whole organism. For example, whether the removal of 50% or 70% of the liver tissue is associated with a greater severity for the test animal as compared to that associated with a mere opening of the abdominal cavity (laparotomy) or complete organ transplantation is not clear. Furthermore, the classification does not take into account the progression of the postoperative

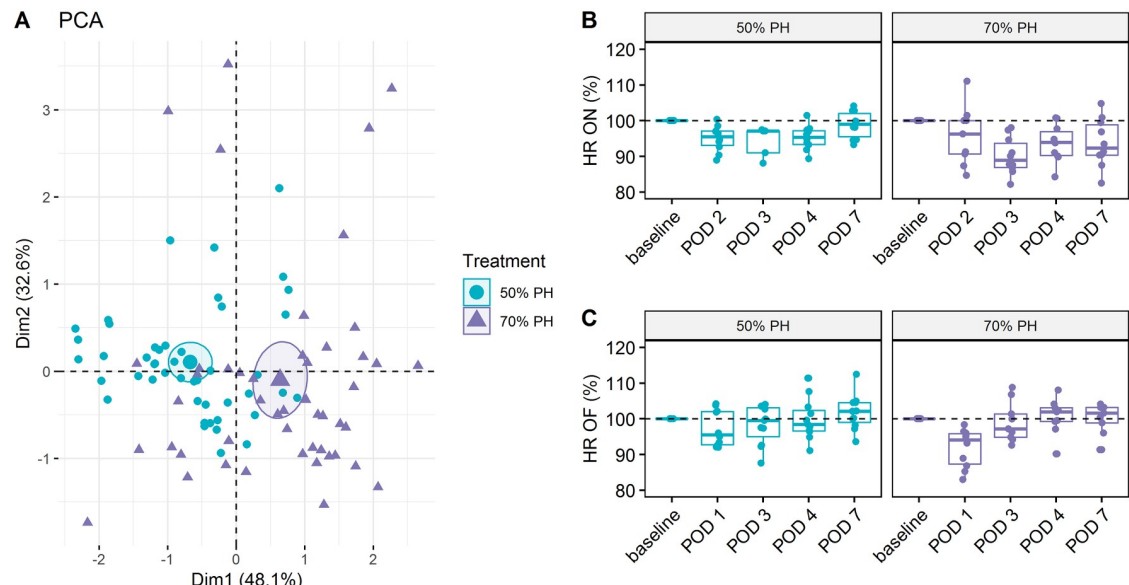

**Fig 11. Principal Component Analysis of telemetric data of 50% and 70% PH groups.** (A) Projection of the two different treatment groups (50% PH and 70% PH) into the two-dimensional PCA factor space; group centroids are characterized by the 95% confidence ellipses. (B) Boxplots of mean heart rates (HR) in overnight measurements for each POD and PH group (50% PH vs. 70% PH); no significant differences between treatments were found. Significant differences of PODs to baseline in the PH groups (Dunnett's test); 50% PH: POD2 vs. baseline $p_{adj}$ = 0.0393; 70% PH: POD3 vs. baseline $p_{adj}$ = 0.0001; POD4 vs. baseline $p_{adj}$ = 0.0177 and POD7 vs. baseline $p_{adj}$ = 0.0195. Note, that the first post-surgical measurement was obtained from POD1 to POD2. (C) Boxplots of heart rates during the Open Field (HR OF) tests for each POD and PH group (50% PH vs. 70% PH); no significant differences between treatments were found. Significant differences of PODs to baseline in the PH groups (Dunnett's test); 70% PH: POD1 vs. baseline $p_{adj}$ < 0.0001.

recovery phase. What is evident, however, is that the severity of an animal that dies unexpectedly during its experiments succumbs to the consequences of its treatments, or reaches humane endpoints must always be classified as "severe" unless the opposite can be verified. For example, studies have shown that, depending on the extent of resection, the effect of liver resection may range from minimal effects to severe impairment of organ function [19–21]. The objective of this study was to assess the extent to which these subtle differences in postoperative severity can be determined using common, objective assessment methods. A further aim was to examine whether it is possible to differentiate between the effect of different surgeries. For this purpose, we investigated various biomarkers, physiological and clinical parameters, and behavioral tests. A novel multimodal approach was used for assessing the influence of severity in three different study groups.

Within the first three days after liver resection, the organism responds to tissue loss [19, 22–24]. During this period, the focus is on restoring the original organ size and function. In rodents, this healing process takes 7–10 days on average, depending on the amount of tissue resected [19, 21, 24]. The results of our study are consistent with these facts. In our study, all animals included achieved and exceeded their baseline body weight by the completion of the study (Fig 3); these findings are consistent with those reported by Fujino et al. in a similar mice model [24]. In addition, evaluation of liver weight (data available in the online repository) demonstrated that all residual liver lobes harvested had regained the initial organ weight on POD7. The surgical procedure and the tissue loss affect the body weight of the animal and blood parameters [24, 25] in addition to causing wound pain [26–29]. While surgical interventions can lead to inflammation and thus to changes in the white blood cell counts, bleeding

that may be caused by liver resection, can lead to changes in the red blood cell counts. These negative influences induce distress and severity for the entire organism [30, 31]. The use of additional objective assessment parameters helped confirm our results. All three test groups exhibited an immediate increase in postoperative severity (severity score, body weight change, OF, and FCMs) with subtile variations. Furthermore, this was supported by the results of AST, ALT, and telemetric data analysis in the PH groups. However, these data are not available for the Sham group to confirm the results. On the evaluation of the individual score parameters, the assessment point "wound healing" showed the highest influence in all groups besides the body weight change. In particular, the independent removal of individual knots from the wound suture with subsequent reddening of the wound caused higher scores on POD4 and POD5 in animals of the Sham group. On the other hand, the removal of sutures within the PH groups sometimes even caused suture dehiscence, which led to higher score points (>10 points) (Fig 4). Although, data of re-sutured animals were included in the analysis, we think that wound healing disorders and surgery-related complications are a physiological bias to a certain extent, which should not be excluded. Even under the impact of these data (9 out of 136 included animals, ≤ 7%), the severity grades were evaluated as mild to moderate.

The OF test is commonly used in the field of neuroscience research; however, its use for assessment of severity based on the voluntary willingness to move after visceral surgery is still quite new. This is reflected in the lack of contemporary literature pertaining to the specific model and setup used in our study. However, this method is often used to test behavioral changes [32, 33] induced by mood-or anxiety-modifying substances [34–36] and emotional state [37]. It is also used to test the intrinsic motivation to flight from predisposed areas despite pain [38–40]. Common to all these disease models is the fact that OF parameters can be used to measure these behavioral changes and thus enable conclusions about severity. This makes it a versatile and sensitive method not only in rat models [41–43].

In a previous study, we reported laparotomy-induced changes in moving activities of rats in the OF test [6]. The results demonstrated that animals show less activity with increasing distress or pain. These findings are consistent with the results shown here (Fig 5). Furthermore, PCA showed that the OF parameters (distance, velocity, and unsupported rearing) were more affected than the other parameters such as body weight, severity score, and FCMs (Fig 7) as well as telemetric data (Fig 10).

The liver plays an important role in corticosterone metabolism. Therefore, the availability of functional liver tissue is a key determinant of FCM level, as the corticosterone metabolites are secreted into the intestine via the bile [15]. The higher FCM values in the Sham group compared to the PH groups may be attributable to the uncompromised liver function in the former (Fig 6). However, the relationship between blood corticosterone and FCM levels is not well characterized. Moreover, the extent to which FCMs are affected by liver function is not clear. Owing to the time delay in the excretion of corticosterone metabolites via the feces, the changes in this parameter are displayed on the following day. Therefore, the surgery-induced high-stress levels of the animals are first reflected in the FCM values at POD1. In contrast, the reduced liver tissue of the PH groups may have resulted in significantly lower FCM values at POD3, POD4, and POD7 (Fig 6A). The standardization of FCM levels based on body weight-to-liver weight ratio of 4.15% allows the comparison of severity levels between laparotomy and liver resection (Fig 6B) on POD1. It is appropriate to adjust this ratio in relation to liver weight proliferation. Thus, liver resection represents a higher severity than laparotomy. Here, it cannot be differentiated unambiguously whether the animals of the PH groups experienced less stress at these times or whether the decreased FCM levels were attributable to reduced liver metabolism and changes in bile acid concentration or its composition [44]. The extent to which these factors affect the measured FCM levels is not clear at present. However, animals of

the Sham group showed no signs of stress at the end of the study. In all experimental groups, levels of both FCMs (Fig 6) and liver-specific enzymes (Figs 8 and 9) on POD7 were close to the respective baseline levels.

Telemetry has been used to determine and evaluate physiological data after surgical interventions [28, 45, 46]. In this study, we used telemetry data pertaining to the 10-minute OF test as well as the activity phase (overnight) (Fig 11). We did not evaluate the internal body temperature as it is liable to be influenced by extraneous factors (subcutaneous position of the device and group housing). In addition, we did not analyze the data pertaining to the night after the abdominal surgery, as the effects of anesthesia may have biased the results. We assessed the influence of liver resection in both groups via the telemetric parameter "HR" during the short OF phase and overnight (Fig 11). No other studies have investigated changes in OF or telemetric data in response to liver resection in rats. In our study, both PH groups showed a reduction in HR in the postoperative phase. An initial increase in HR in response to the intraoperative blood loss and the drop in blood pressure should be expected. In addition, increased HR is also expected in the further postoperative course due to wound healing and handling stress. However, this increase was not evident in both groups until POD3. Only the HRs during OF were above the baseline levels until the end of the study, in contrast to overnight levels. This leads to the assumption that, on average, the animals remain hypoactive overnight, which resulted in a lower heart rate as compared to the baseline level. In summary, however, the results of PCA showed that the OF parameters "distance" and "unsupported rearing" provided more additional information compared to telemetric data or body weight (Fig 10). In the first dimension, the OF parameter "distance and unsupported rearing" showed a greater influence on the overall variance than the body weight. While the parameters of telemetry and OF represent short-term changes in the OF test, the body weight does not change much during this time. Therefore, its contribution to the variance is low. Beyond this, we did not include the additional OF parameter "velocity" in the PCA, because of its similarity to "distance" (Fig 5). Rather, we believe that unsupported rearing is affected by abdominal surgery and, therefore, it provides an additional perspective to stress assessment.

## Conclusion

The prognostic severity grading of this study was classified as "moderate" according to Annex VIII of the EU Directive 2010/63. However, based on our findings, a subtle distinction can be made between the surgical procedures performed. The parameters body weight and severity score only indicated mild to moderate severity in the Sham and PH50 groups, which recovered after 3–4 days. However, in the PH70 group, the exposure appeared to be "moderate" and the animals required a longer time to reach the initial values (POD5 and POD6). The additionally evaluated objective assessment parameters (FCMs, OF distance, velocity and unsupported rearing, HR, blood pressure) indicate that the exposures after Sham and 50% PH require up to 7 days of reconvalescence; however, for the 70% PH group, the assessment period was not sufficient to return to basal values. In this case, a longer reconvalescence period is recommended. Furthermore, we were able to demonstrate that the OF test is the most sensitive method to assess severity in the animals, within 10 minutes of observation time. However, the extent to which the choice of the rat strain and the sex of the animals influenced the severity is open to question. In addition, future studies should compare the severity of organ transplantation with that of the models presented here, although all these interventions are rated as "moderate". Nevertheless, for the animals that died due to unknown reasons before the end of the study, the severity grade still has to be classified as "severe".

Further, the effect of change in analgesic regimen on postoperative severity should also be investigated. The development of reliable methods for the assessment of the long-term welfare of laboratory animals is a key imperative. Identification of valid, model-specific parameters that can determine individual severity levels in the animal is an important step to achieve this goal. The use of multimodal, objective parameters that can be measured with the least invasive methods offer a distinct advantage in this respect.

## Supporting information

**S1 File. Score sheet.** Experiment-specific score sheet for daily severity assessment according to Morton DB et al.
(PDF)

**S1 Checklist.**
(PDF)

## Acknowledgments

The authors are grateful to Mareike Schulz, Pascal Paschenda, Stefan Bruch, and Dr. Yuuki Masano for skillful technical assistance and Edith Klobetz-Rassam for FCM analysis.

## Author Contributions

**Conceptualization:** Leonie Zieglowski, René H. Tolba.

**Data curation:** Leonie Zieglowski, Anna Maria Kümmecke, Steven R. Talbot.

**Formal analysis:** Leonie Zieglowski.

**Funding acquisition:** René H. Tolba.

**Investigation:** Leonie Zieglowski.

**Methodology:** Leonie Zieglowski, René H. Tolba.

**Software:** Steven R. Talbot.

**Supervision:** René H. Tolba.

**Validation:** Leonie Zieglowski.

**Visualization:** Leonie Zieglowski.

**Writing – original draft:** Leonie Zieglowski.

**Writing – review & editing:** Anna Maria Kümmecke, Lisa Ernst, Rupert Palme, Ralf Weiskirchen, Steven R. Talbot, René H. Tolba.

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
