## [Decision Letter · Decision Letter 0]

20 Apr 2021

PONE-D-21-06410

Assessing the severity of laparotomy and partial hepatectomy in male rats – a multimodal approach

PLOS ONE

Dear Dr. Zieglowski,

Thank you for submitting your manuscript to PLOS ONE. After careful consideration, we feel that it has merit but does not fully meet PLOS ONE’s publication criteria as it currently stands. Therefore, we invite you to submit a revised version of the manuscript that addresses the points raised during the review process.

As you can see, both reviewers appreciated your work and suggested only to add some information and to clarify a couple of things.  

We look forward to receiving your revised manuscript.

Kind regards,

Pavel Strnad

Academic Editor

PLOS ONE

Journal Requirements:

Additional Editor Comments (if provided):

Reviewers' comments:

Reviewer's Responses to Questions

**Comments to the Author**

1. Is the manuscript technically sound, and do the data support the conclusions?

Reviewer #1: Yes

Reviewer #2: Yes

2. Has the statistical analysis been performed appropriately and rigorously? 

Reviewer #1: Yes

Reviewer #2: Yes

3. Have the authors made all data underlying the findings in their manuscript fully available?

Reviewer #1: Yes

Reviewer #2: Yes

4. Is the manuscript presented in an intelligible fashion and written in standard English?

Reviewer #1: Yes

Reviewer #2: Yes

5. Review Comments to the Author

Reviewer #1: The authors present very interesting data on the highly important subject of severity assessment in animal research. This topic frequently is based on subjective criteria and empirical knowledge. However, authorities, animal welfare officers or designated veterinarians as well as researchers need scientific-based methods to assess severity in different animal models. This paper significantly contributes to the literature on this subject.

The authors should address the following minor points of criticism in order to enable to readers to fully interpret the presented results.

Abstract:

The authors are summarizing the heart rate (HR) and blood pressure (BP) data as telemetric data. However, it is also possible to determine activity by telemetry. They should therefore specify in relevant parts of their manuscript (like abstract) which data (HR and BP) have been determined telemetrically.

Introduction:

Line 39: The authors should mention that severity categorization has a prospective and retrospective component.

Fig 1: the authors should re-consider whether Fig 1 is necessary at all. It does not contribute to the manuscript in as significant way.

Line 71 to 75: The authors state that “behavioral testing was used to evaluate the distress, pain, and the affective internal state” while also saying that “Here, the OF test was used to evaluate the spontaneous locomotor behavior and voluntary willingness to move…”. These two statements sound contradictory as assessment of distress, pain and affective state regularly contain far more parameters than only parameters of activity. This should be rephrased.

Material and Methods:

Lines 83 to 86 and lines 101 to 104 contain the same contents. Please delete one version.

Line 126: authors state in legend of Fig 2. that 4 animals per PH group were implanted with transmitters. In Table 1. it says 12 animals. Please clarify and correct if necessary.

Line 133: Authors state that postoperative scoring is done 3 times a day from POD 1 to POD3. This raises the question about the resulting data as presented in Fig. 5. Do the data as presented for POD1, POD2 and POD3 in Fig 5 represent a single daily observation or mean values of all three observations.

In the OF test is unclear which measurements have been used as baseline? In line 257 it is stated that the values of D18 and D19 have been averaged for baseline. However, this obviously is an error. We assume that the authors probably meant days D-18 and D-19. If so, the baseline value D-19 must be included in Fig 2. Have the animals been filmed from above for OF? This important information should be added. Further, the authors should explain why they have chosen unsupported rearing in contrast to supported rearing? Have they expected this parameter to have a special value for their results? They should address this briefly in the discussion.

Severity Scoring: The modified severity scoring should be described in more detail. It is unclear how this scoring was composed exactly and which score (in points) was given for the various findings. Maybe there was a scoresheet, which could be added to the manuscript as supplementary material? It is for example unclear, what was included in assessment of “4) surgical procedure and wound healing” and which scores have been assigned to a bleeding wound/dehiscence. It is further of relevance, where scoring took place (in the home cage?).

Line 207: The authors should describe in the “Materials Section” which body weights were used as base line. In line 271 of the “Results Section” it is mentioned that the post-surgical body weights have been used for this purpose. In this context we also ask ourselves why the authors did not use the pre-surgical body weights as baseline. This would have excluded the possibility that the body weights had been influenced by surgery.

Line 209: Did single suture dehiscence occur? If so, how often and how was this handled? Was additional anesthesia necessary in these cases?

Blood parameters: Could the authors explain why blood parameters were assessed at all? What was the background for using these parameters?

FCMs: The authors only present fecal corticosterone metabolites in their study. However, in line 220 they also mention that corticosterone had been determined in the serum. Could the authors please check this.

Telemetric data: In line 236 the abbreviation “HR” is mentioned for the first time without providing the complete term. Further, the authors should explain in which intervals/bins the blood pressure was assessed during telemetric assessment.

In Line 253 the abbreviation “PCA” is mentioned for the first time, but without providing the complete term.,

According to line 260 the samples which have been collected during second surgery (PH or sham) have been used as baseline of FCMs. However, in Fig 7 the authors present data for a “baseline” and “surgery” value. The authors should make clear which value has been used as baseline.

The sentence part in line 343 "were influenced by variance" does not make sense, neither does "Therefore, these are relevant in all three study groups" in line 345. The authors should explain/reformulate. By our understanding, the variance explanation percentage of the factors only measures how many variables can be represented by each factor and how well. It does not measure how relevant the contributing variables are. You can however conclude how relevant the different factors (i.e. all contributing variables together) are in regard to differentiating between groups by observing the actual graphs.

In general: Have the researchers been blinded to the animal’s treatment? This seems especially important for severity scoring.

Results:

In general: The authors should try to show significant p values for group differences as outlined in the legends of Figures 4, 5, 7, 9, and 10 in the respective Figures themselves. This would help the reader to understand the complex data.

In figures 8, 10, and 11 the authors plot two PCA dimensions each, but only show which variables contribute to dimension 1. Which variables contribute to the second one?

Animal/sample numbers included in each parameter should be stated in the respective legend.

In line 247 the authors mention that statistical anlyses have been performed to compare their data with the baselines. However, no results of these analyses are presented in the manuscript.

Line 286: I would suggest to write “end of study” instead of “endpoint”. The difference between study end and humane endpoint becomes clearer this way.

Line 291: The authors describe that 2 animals of the 70% PH group died in the nights between POD1 and POD3 for unknown reasons. The EU working document on a severity assessment framework reads „The actual severity for animals found dead should be reported as 'severe' unless an informed decision can be made that the animal did not experience severe suffering prior to death“. The authors should discuss their decision to categorize PH70% as a manipulation of moderate severity (which is indeed supported by the reviewers) under the aspect of the two animals which were found dead, in particular as the cause of death was unknown. Was data from these animals excluded from the study?

Legend of Fig 7B: It should be mentioned that the data shown refer to POD1.

Fig. 8: What is the dashed line in Fig 8B? This should be described in all Figures with PCA containing such a line.

Discussion:

Line 457f: The authors state that “all three groups exhibited an immediate increase in postoperative severity”. This statement should be re-considered for the following reasons:

o No telemetric data are available for the sham group

o No ALT and AST values are available for the sham group for the critical test day POD1.

o The severity score of the sham group does not seem to be increased on POD1, POD2, and POD3.

Line 459: The results on severity scoring discussed in this paragraph are hard to interpret as the composition of the score is not described properly before. If the score is influenced heavily by wound healing parameters, one must ask if the resulting scoring really justifies to draw conclusions about mild/moderate/severe severity from that score as found in figure 5. Maybe the authors could discuss if the used severity score in fact presents overall burden on the animal due to general anesthesia and surgery or if the big influence of the category wound healing is leading to a wrong classification in higher severity grades.

Line 524f: Could the authors explain why they consider the unsupported rearing as additional perspective to stress assessment? They mentioned that it is affected by abdominal surgery meaning that it is decreased post op. Is it therefore not rather a parameter potentially indicating abdominal pain or discomfort? Do the authors know of results in rearing behavior in anesthetized/untreated rats to distinguish between the effects of pain vs. effects of anesthesia or handling? One could ask if the applied analgesic regimen is sufficient for the described laparotomy. Maybe the authors could briefly discuss this point.

Reviewer #2: The article written by Zieglowski et al. tackles the question of surgical severity following partial hepatectomy in rats. It is clear and well written. Here are a few comments and questions raised that need to be answered in order to improve the quality of the paper and make it accepted for publication.

* the use of metamizole as analgesics is good for visceral considerations but does not sound to be sufficient. Why not using opioids in order to improve analgesia ? Moreover length of analgesia by metamizole is short since only one administration is performed (or until POD3, not clear, line 190 ?), which means there is no more pain after 12-24 hours. Why not carrying on 2 to 3 days post-surgery ?

* why are only male rats used ? What is the explanation ?

* is it really useful to use antbiotics for such a surgery ? If yes, is it really useful to use a 2nd/3rd generation of antibiotics ?

* line 177: could you precise the exact percentage of liver withdrawn ? LLL + CL + ML make 78 % (not 70 %). Same for 50 %: it is not clear.

* line 458: typo (subtle)

* why do you use FCM and not blood corticoide levels ? the explanation given in the discussion should be clarified.

* could you precise the exact interest of telemetry in your experiences ?

6. PLOS authors have the option to publish the peer review history of their article (what does this mean?). If published, this will include your full peer review and any attached files.

Reviewer #1: No

Reviewer #2: **Yes: **Dr. Thomas HUBERT, Ass Prof in Surgery, DVM, PhD, Dip Vet LAS

---

## [Author Response · Author response to Decision Letter 0]

2 Jun 2021

Dear Madam/Sir:

First of all we would like to thank the Editors and the Reviewers for their valuable input and their pro bono work.

We tried to meet the suggestions of the reviewers and hope that this improved the quality of our manuscript, accordingly. In addition, our response to the individual comments of the reviewers are provided in this point-to-point response letter.

Referee(s)' Comments to Author:

Reviewer #1:

Abstract: The authors are summarizing the heart rate (HR) and blood pressure (BP) data as telemetric data. However, it is also possible to determine activity by telemetry. They should therefore specify in relevant parts of their manuscript (like abstract) which data (HR and BP) have been determined telemetrically.

We thank you for addressing this point and we revised the text. We hope that the additional information, especially in the abstract, contributes to specify the relevant parameters and approaches of the manuscript.

Please see the revised abstract in line 26 – 31:

“… Post-surgical severity assessment was performed via several multimodal assessment tools: I) model-specific score sheet focusing on body weight, general condition, spontaneous behavior and the animals willingness to move as well as on wound healing; II) Open Field tests evaluating the total distance and velocity an animal moved within 10 minutes and its rearing behavior during the test; III) telemetric data analysing heart rate and blood pressure; and IV) analysis of blood (AST, ALT and hemogram) and fecal samples (fecal corticosterone metabolites). …”

Introduction: Line 39: The authors should mention that severity categorization has a prospective and retrospective component.

We thank you for this supplementary aspect and have added it to line 44 of the revised manuscript.

“All scientific research projects are currently required to categorize the expected and the retrospective actual degree of severity … ”

Fig 1: the authors should re-consider whether Fig 1 is necessary at all. It does not contribute to the manuscript in as significant way.

We thank you for this constructive advice and are glad that we were able to describe the interaction of the multimodal approach of severity assessment sufficiently already in the text. We have therefore deleted Figure 1 from the manuscript and lined up all other figures.

Line 71 to 75: The authors state that “behavioral testing was used to evaluate the distress, pain, and the affective internal state” while also saying that “Here, the OF test was used to evaluate the spontaneous locomotor behavior and voluntary willingness to move…”. These two statements sound contradictory as assessment of distress, pain and affective state regularly contain far more parameters than only parameters of activity. This should be rephrased.

You are right, that the test method “Open Field” alone is not able to detect and measure "distress", "pain" and "affective internal state" in rats. Moreover, this test is one of our multimodal approach for severity assessment in this study (telemetry, blood parameter measurement, FCMs, etc.). Here, our approach is to determine whether these negative impact factors (e.g., laparotomy, partial hepatectomy) also have an influence on the spontaneous willingness to move and the spontaneous locomotor behaviour of the animals in their Open Field performance. Therefore, we tried to determine, measure and also evaluate the assessed parameters of this test method, according to the results of our pilot study , in this more invasive experiment.

To describe this clear, we deleted the first part of this statement (former line 71). The revised part of the text reads as follows (line 77 -82):

“… As a further method, behavioral testing via the OF test was used to assess postoperative severity in this surgical model involving the use of living animals as described elsewhere [3]. Here, the focus was set on the evaluation of spontaneous locomotor behavior and voluntary willingness to move as indices of surgery-related severity in the animal.”

Material and Methods: Lines 83 to 86 and lines 101 to 104 contain the same contents. Please delete one version.

Thank you for your comment. As suggested, we deleted the redundant section in lines 108 - 111 in the revised manuscript.

Line 126: authors state in legend of Fig 2. that 4 animals per PH group were implanted with transmitters. In Table 1. it says 12 animals. Please clarify and correct if necessary.

We thank you for the corrective note. We have revised the number of animals in the legend of figure 1 (former Fig 2). The number of animals from each 7-day survival group is still n = 12.

Line 133: Authors state that postoperative scoring is done 3 times a day from POD 1 to POD3. This raises the question about the resulting data as presented in Fig. 5. Do the data as presented for POD1, POD2 and POD3 in Fig 5 represent a single daily observation or mean values of all three observations.

We apologize for the misleading description. Yes, due to the more frequent examinations within the first three post-surgical days (3 scorings per day for POD 1 - POD 3), these boxplots of figure 4 (former figure 5) contain all the corresponding individual values. Same was done with the single values of the following days scores (POD 4 - POD 7), with only one scoring per day. Therefore, the boxplots do not show mean values, but the distribution of the individual values and the corresponding median of each POD.

To clarify this, we have added a supplementary paragraph in line 231 - 232 of the revised manuscript:

“For statistical analysis, the individual scores on each POD were shown as boxplots.”

In the OF test is unclear which measurements have been used as baseline? In line 257 it is stated that the values of D18 and D19 have been averaged for baseline. However, this obviously is an error. We assume that the authors probably meant days D-18 and D-19. If so, the baseline value D-19 must be included in Fig 2. Have the animals been filmed from above for OF? This important information should be added. Further, the authors should explain why they have chosen unsupported rearing in contrast to supported rearing? Have they expected this parameter to have a special value for their results? They should address this briefly in the discussion.

We thank you for this constructive advice and apologize for the unclear statement. To clarify, it must be said that the baseline of our study is the second repetition (2nd re-training) on day D-1. Here, D18 and D19 do not exist because the study only runs up to D7 (= POD7). Day D-18 corresponds to the 2nd training (2nd re-training) and since the animals are still habituating at this point, this was not set as baseline. We have therefore corrected the text and changed it to D-2 and D-1 to make it clearer that the re-trainings (prior to abdominal surgery) are the baseline. Please see orange ring in figure 1 (former figure 2), here, as an aid for better visualization:

Therefore, please refer to line 285 of the revised manuscript.

We added the information, that the video tracking was performed from above in line 150 of the revised manuscript.

Yes, we hypothesized that due to the laparotomy of the surgery, the animals might have more difficulties or even pain during unsupported rearing than with assistance of the Open Fields wall sides. This can be observed in patients in human medicine, too, as the abdominal muscle layer is impaired by the surgery leading to avoidance of unsupported rising.

The analysis of supported and unsupported rearing behavior in our study showed, that the supported rearing behavior was less affected due to the abdominal surgery, than the unsupported rearing. Again, we made the comparison between the PH groups and the SHAM group. The normalization of the depicted values to % was based on animal individual differences within the groups.

Although there were no significant differences in unsupported rearing between the PH and SHAM groups at the different PODs (figure 6 of first manuscript version); they were found within the groups compared to their baselines. This allows the assumption that the animals are impaired by the abdominal surgery and that the frequency of unsupported rearing allows conclusions regarding existing severity. Although, in the supported rearing behavior significant changes occurred within the PH50% (POD4) and PH70% (POD1) group (see figure 1 below), the total amount of significances is higher in comparison of unsupported rearing behavior (see figure 2 below). Although in none of the comparisons (figures 1 and 2) the SHAM group showed significant changes to its baselines, however, in figure 2 the PH groups differ several times and significantly from their own baseline values (PH50%: POD1, 3 and 7; PH70%: POD 1).

Figure 1: Comparison of supported rearing behavior (%) in various groups (Sham; 50% PH; 70% PH). Dunnett´s test was used for comparison with baseline; animal numbers of SHAM (baseline to POD 7= 10 rats); animal numbers of PH50% (baseline and POD 1=63, POD 3= 42, POD 4 and POD 7= 21 rats); animal numbers of P70% (baseline= 63, POD 1= 58, POD 3= 37, POD 4 and POD 7= 18 rats).

Figure 2: Comparison of unsupported rearing behavior (%) in various groups (Sham; 50% PH; 70% PH). Dunnett´s test was used for comparison with baseline; animal numbers of SHAM (baseline to POD 7= 10 rats); animal numbers of PH50% (baseline and POD 1=63, POD 3= 42, POD 4 and POD 7= 21 rats); animal numbers of PH70% (baseline= 63, POD 1= 58, POD 3= 37, POD 4 and POD 7= 18 rats).

Severity Scoring: The modified severity scoring should be described in more detail. It is unclear how this scoring was composed exactly and which score (in points) was given for the various findings. Maybe there was a scoresheet, which could be added to the manuscript as supplementary material? It is for example unclear, what was included in assessment of “4) surgical procedure and wound healing” and which scores have been assigned to a bleeding wound/dehiscence. It is further of relevance, where scoring took place (in the home cage?).

We are delighted to comply with your request and have enclosed the score sheet as supplementary material to the manuscript.

Where possible, scoring was carried out first in the home cage (to observe and evaluate the animal without further “external” disturbance) and then by the scorer outside the cage by removal and examination. To clarify this in the text, we added this in line 217 - 218 of the revised manuscript:

“The scoring was carried out in the home cage first, without further interference from outside, hereafter each animal was examined in detail, outside the cage. …”

Line 207: The authors should describe in the “Materials Section” which body weights were used as base line. In line 271 of the “Results Section” it is mentioned that the post-surgical body weights have been used for this purpose. In this context we also ask ourselves why the authors did not use the pre-surgical body weights as baseline. This would have excluded the possibility that the body weights had been influenced by surgery.

We would like to apologize for our unclear wording.

As stated in line 95, animals in an age class of 6-8 weeks and a body weight of 150 – 175 g were purchased for this study from a professional breeder. However, this time of delivery is at least 28 days before the day of liver resection, so the animals will have gained body weight by then. The body weight at the time of the liver resections was between 258g - 415g.

The reason why we used the post-surgical body weight as the baseline value is, that we do know, that the surgery influenced body weights. By resecting 50% or 70% of the liver mass, the body weight of the animals has been reduced by an average of 6.8 g (in 50%PH) or 8.01 g (in 70%PH) liver weight. In order not to assess this bias negatively for the animals in the course of the post-surgical scoring, the weight measured right after the surgery (post-surgical) was set as the baseline value.

Line 209: Did single suture dehiscence occur? If so, how often and how was this handled? Was additional anesthesia necessary in these cases?

Yes, it happened that in animals single stitches of their abdominal sutures opened, but only in the Sham and 70% resection groups.

Here: 1 abdominal suture had to be fixed with wound clips (PH70%); 1 suture had to be re-sutured under general anaesthesia (SHAM group); 6 animals got a hernia of the muscle layer and had to be re-sutured under general anaesthesia (PH70%) out of whom 1 animal had to get its skin suture fixed with a wound clip, afterwards.

Basically, this is how we proceeded: If only the sutures were gnawed, but the wound edges were adapted, nothing was done. If the adaptation of the wound edge was about to loosen, we put a wound clip over the corresponding area to secure it, which was removed again after a few days (as soon as the wound looked healthy).

If this time point could not be determined properly, for example at night, and the skin suture was already open at the next scoring time due to the lack of single knots, the animal was again put under short isoflurane anaesthesia, the wound was cleaned, and re-sutured.

In the case of ruptures of the abdominal muscle layer, but with the skin suture closed, the animal was also put under inhalation anaesthesia, the skin suture re-opened, the abdominal muscles re-sutured and the skin closed as usual.

If this occurred outside the time window covered with painkillers and antibiotics, this was injected accordingly to comply with the protocol.

Blood parameters: Could the authors explain why blood parameters were assessed at all? What was the background for using these parameters? It is known that surgery in general and liver resection in particular, as well as analgesia (metamizole) can have effects on individual blood parameters. While surgical interventions can lead to inflammation and thus to changes in the white blood cell counts, bleeding that may be caused by liver resection can lead to changes in the red blood cell counts.

On the other hand, metamizole is known to cause agranulocytosis in patients 1, which we could not confirm in our animals. In order to identify and verify such changes, which could have an impact on severity grades, the blood cell counts were assessed in this study and presented in the manuscript to complete the data.

FCMs: The authors only present fecal corticosterone metabolites in their study. However, in line 220 they also mention that corticosterone had been determined in the serum. Could the authors please check this. You are right that this data is not presented in the manuscript. This is due to the fact that we had very high expectations for this parameter in the experimental design, but it turned out to be too unreliable for evaluation due to the very short half-life in serum. While significant changes in the FCMs were visible, the blood corticosterone values were often below the measurable threshold (6.1 ng/ml) at the time of the blood samples or the variation was too wide. Therefore, no valid data sets with statistical significance could be generated. For the sake of completeness, however, this was indicated in the text, as the quantities of blood taken must be taken into account as possible factors influencing severity grades.

Telemetric data: In line 236 the abbreviation “HR” is mentioned for the first time without providing the complete term. Further, the authors should explain in which intervals/bins the blood pressure was assessed during telemetric assessment.

We apologize for the incorrect term. We added the missing information in the sentence. Please see the revised version of the manuscript, line 258-260:

“… The parameters evaluated were heart rate (HR) and blood pressure (BP) and values of each measurement time span (OF or overnight) were averaged with a logging rate of 1 minute.”

In Line 253 the abbreviation “PCA” is mentioned for the first time, but without providing the complete term. Here again, we apologize the missing information. We completed the text as followed. Please refer to line 280 - 281 of the revised manuscript:

“ A Principal Component Analysis (PCA) was performed …”

According to line 260 the samples which have been collected during second surgery (PH or sham) have been used as baseline of FCMs. However, in Fig 7 the authors present data for a “baseline” and “surgery” value. The authors should make clear which value has been used as baseline.

We thank you for this helpful comment and apologize for the unclear wording. At this point, FCMs and blood levels are not identical, as they have different baselines. We have therefore revised the sentence and adjusted it accordingly (see lines 287 – 290).

“Baseline values for blood and serum analysis were averaged out of samples obtained during second surgery (Sham or PH). For FCM analysis, values of weaned feces from the second retraining were averaged and used as baselines. …”

The sentence part in line 343 "were influenced by variance" does not make sense, neither does "Therefore, these are relevant in all three study groups" in line 345. The authors should explain/reformulate. By our understanding, the variance explanation percentage of the factors only measures how many variables can be represented by each factor and how well. It does not measure how relevant the contributing variables are. You can however conclude how relevant the different factors (i.e. all contributing variables together) are in regard to differentiating between groups by observing the actual graphs. We performed a Principal Component Analysis (PCA) to investigate how well the above-mentioned parameters (body weight, severity score, OF distance, velocity, unsupported rearing, and FCMs) were represented in, e.g., the first two dimensions. The three OF parameters were found to be the main contributors to the total variance representation of the first dimension (46.1%). Here, the magnitudes of the corresponding values in the eigenvectors indicated that in regard of differentiating between the three treatment groups, they can be considered as the most relevant within that dimension.

In general: Have the researchers been blinded to the animal’s treatment? This seems especially important for severity scoring.

We would like to thank the reviewer to raise this important point. However, due to the size of the working group and the number of surgeries we performed for this study, we were not able to perform the scoring with additional researchers, trained and experienced enough for scoring, blinded to the treatment.

According to our own unpublished data, we analyzed severity scoring as well as the interrater variability of different severity grades of rodents via pictures. The data showed no significances in the evaluation of none or severe severity grades within an inexperienced or experienced rater. Only pictures of animals with mild to moderate experimental affection, seemed to be difficult to rate and to assign the moderate severity grade.

We therefore added a sentence to clarify the possibility of a bias resulting from this issue. Please refer to line 214 – 217 in the revised version:

“…Due to the number of surgeries performed, the member size of the working group and the additional limitation of relying on female-only staff with sufficient expertise, the personnel was not blinded to the animals treatment. …”

Results: In general: The authors should try to show significant p values for group differences as outlined in the legends of Figures 4, 5, 7, 9, and 10 in the respective Figures themselves. This would help the reader to understand the complex data.

We tried to meet your suggestions and included both the significant p values as well as the animal/sample numbers of each parameter in the legends of the figures. We hope, that this does not impair the readability and clarity of the figures.

Here, the following symbols represents the significances between the groups:

# : Sham vs. 50% PH

* : Sham vs. 70% PH

& : 50% vs. 70% PH

Moreover, the quantity of symbols corresponds to the level of significance: # / */ & = p ≤ 0.05 up to ####/ ****/ &&&& = p ≤ 0.0001.

We further added this additional information in the text of the revised manuscript.

See line: 275 – 278.

Please see the following revised Figure 4 (now figure 3):

Fig 3. Comparative body weight change (%) in the three study groups (Sham, 50% PH, 70% PH) during post-surgical phase; animal numbers (Sham/ 50% PH / 70% PH): post-surgery (n = 10/63/59), POD 1 (n = 10/63/59), POD 2 (n = 10/42/39), POD 3 (n = 10/42/38), POD 4 (n = 10/21/18), POD 5 (n = 10/21/18), POD 6 (n = 10/21/18), POD 7 (n = 10/21/18); two-way ANOVA F(14,621) = 4.026, p < 0.0001; POD2: Sham vs. 50% PH (#) padj = 0.0125 and Sham vs. 70% PH (***) padj = 0.0005; POD3: Sham vs. 50% PH (###) padj = 0.0002 and Sham vs. 70% PH (****) padj = < 0.0001; POD4: Sham vs. 70% PH (*) padj = 0.03 and 50% PH vs. 70% PH (&) padj = 0.0313; POD5: Sham vs. 70% PH (**) padj = 0.0072 and 50% PH vs. 70% PH (&&) padj = 0.0011; POD6: Sham vs. 70% PH (*) padj = 0.0105 and 50% PH vs. 70% PH (&&&) padj = 0.0001; POD7: Sham vs. 50% PH (#) padj = 0.0250 and 50% PH vs. 70% PH (&&&&) padj < 0.0001.

Please see the following revised Figure 5 (now figure 4):

Fig 4. Boxplots of severity scores (body weight, general condition, wound healing, and spontaneous locomotor behavior) after Sham or PH with upper limits with a gradual allocation of severity (mild: ≥ 5 points, moderate: ≥ 10 points, and severe: ≥ 20 points); animal numbers (Sham/ 50% PH / 70% PH): POD 1 (n = 10/63/60), POD 2 (n = 10/42/39), POD 3 (n = 10/42/38), POD 4 (n = 10/21/18), POD 5 (n = 10/21/18), POD 6 (n = 10/21/18), POD 7 (n = 10/21/18); Kruskal–Wallis-test (χ2 = 353.3, p < 0.0001, degrees of freedom (df) = 21); POD1: Sham vs. 50% PH (####) and Sham vs. 70% PH (****) padj < 0.0001; POD2: Sham vs. 50% PH (####) and Sham vs. 70% PH (****) padj < 0.0001; POD3: Sham vs. 50% PH (#) padj = 0.0253 and Sham vs. 70% PH (****) padj < 0.0001.

Please see the following revised Figure 7 (now figure 6):

Fig 6. Comparison of fecal corticosterone metabolites (FCMs) in samples weaned during Open Field or during anesthetic induction in various groups. (A) Percentage changes in FCM levels in the three groups over time; sample sizes (Sham/ 50% PH / 70% PH) on each time point: baseline (n = 10/61/62), surgery (n = 10/54/57), POD 1 (n = 9/54/55), POD 3 (n = 10/40/36), POD 4 (n = 10/20/18), POD 7 (n = 10/19/18); F(10,528) = 2.062, p < 0.0001; surgery: 50% PH vs. 70% PH (&&) padj = 0.0031; POD1: Sham vs. 70% PH (*) padj = 0.0377 and 50% PH vs. 70% PH (&) padj = 0.0299; POD3: Sham vs. 50% PH (#) padj = 0.0169; Sham vs. 70% PH (****) padj < 0.0001 and 50% PH vs. 70% PH (&) padj = 0.0273; POD4: Sham vs. 50% PH (#) padj = 0.0137 and Sham vs. 70% PH (****) padj < 0.0001. (B) Standardization of FCMs levels (μg)/g feces in relation to liver weight (calculated by BW-LW-ratio) at POD1; sample sizes: Sham n = 9, 50% PH n = 54, 70% PH n = 55. For liver weight, a value of 4.15% of total body weight was calculated (own data). Data were checked for normality and homoscedasticity using the Shapiro–Wilk and the Levene test. Owing to the observed variance between groups (p = 0.007), the non-parametric Kruskal-Wallis test was used (χ2 = 20.9, df = 2, p < 0.0001). The observed between-group differences were analyzed using the Dunn's post-hoc test and the false-discovery rate for multiple tests was controlled using the Benjamini-Hochberg criterion. Significant differences were found between the 50% and the 70% PH group (Z = -3.727, padj = 0.0006) as well as between the 70% PH and the Sham group (Z = 3.541, padj = 0.0006).

Please see the following revised Figure 9 A (now figure 8 A):

Please see the following revised Figure 9 B (now figure 8 B):

Fig 8. Serum measurements. Boxplots of serum (A) alanine aminotransferase (ALT) and (B) aspartate aminotransferase (AST) levels (U/L) showing group-individual values in group comparisons (Sham, 50% PH, 70% PH); Sham: TI (n = 9), laparotomy (n = 10), eutha POD 7 (n = 10); 50% PH: TI (n = 12), liver resection (n = 63), eutha POD 1 (n = 21), eutha POD 3 (n = 21), eutha POD 7 (n = 21); 70% PH: TI (n = 12), liver resection (n = 61), eutha POD 1 (n = 18), eutha POD 3 (n = 20), eutha POD 7 (n = 18); ALT: one-way ANOVA F(12,277) = 54.71, p < 0.0001; POD1: 50% PH vs. 70% PH (&&&&) padj < 0.0001. AST: one-way ANOVA F(12,276) = 26.05, p < 0.0001; POD1: 50% PH vs. 70% PH (&&&&) padj < 0.0001.

As a PCA, original figure 10 (now figure 9) does not contain any significances that could be transferred to the graph. Therefore, this figure remains unchanged.

In figures 8, 10, and 11 the authors plot two PCA dimensions each, but only show which variables contribute to dimension 1. Which variables contribute to the second one?

We have adjusted the graphics according to your comments and added data for the second dimension. Please see revised figures 8, 10 and 11 as followed:

Line 383 – 387 of the revised manuscript (former figure 8):

“Fig 7. Principal Component Analysis (PCA) of the Sham, 50% and 70% PH groups. (A) Projection of individual treatment groups into the two-dimensional PCA factor space; group centroids are characterized by the 95% confidence ellipses. (B) Variance contributions of factors in percentage in the first and (C) second dimension. BW, body weight; FCMs, fecal corticosterone metabolites (μg/g); OF, Open Field; OF RearingUnsupp, rearing unsupported; dashed line shows the cut-off for the uniform variance distribution.”

Line 419 – 426 of the revised manuscript (former figure 10):

“Fig 9. Principal Component Analysis of blood and serum parameters of 50% and 70% PH groups. (A) Projection of individual euthanasia time-points (POD1, POD3. or POD7) into the two-dimensional PCA factor space; group centroids are characterized by the 95% confidence ellipses. (B) Projection of the two different treatment groups (50% PH and 70% PH) into the two-dimensional PCA factor space; group centroids are characterized by the 95% confidence ellipses. (C) Variance contributions of factors in % in the first and (D) second dimension. AST, alanine aminotransferase; ALT, aspartate aminotransferase; Baso, basophils; BW, body weight; Eos, eosinophils; Gluc, glucose; HGB, hemoglobin; LDH, lactate dehydrogenase; Leuko, leukocyte count; PLT, platelets; RBC, red blood cell counts; dashed line shows the cut-off for the uniform variance distribution.”

Line 441 – 449 of the revised manuscript (former figure 11):

“Fig 10. Principal Component Analysis of telemetric data including body weight, telemetric data, and OF parameters in the 50% PH and 70% PH groups. (A) Projection of different test time-points (baseline, POD1, POD3, POD4, and POD7) of 50% PH group into the two-dimensional PCA factor space; group centroids are characterized by the 95% confidence ellipses. Variance contributions of factors (A) in % in the first (B) and second (C) dimension. (D) Projection of different test time-points (baseline, POD1, POD3, POD4, and POD7) of 70% PH group into the two-dimensional PCA factor space; group centroids are characterized by the 95% confidence ellipses. Variance contributions of factors (C) in % in the first (E) and second (F) dimension. BP, blood pressure; BW, body weight; HR, heart rate; OF, Open Field; dashed line shows the cut-off for the uniform variance distribution.”

Animal/sample numbers included in each parameter should be stated in the respective legend. We comply with your request and have enclosed the missing information about animal numbers and sample sizes in the legend of the respective graphs/figures. To show this as consistently as possible, these are shown in the format "n = Sham/ 50% PH/ 70% PH”.

Therefore, please see revised legends of the figures 4, 5, 7 and 9 :

Former Figure 4:

Fig 3. Comparative body weight change (%) in the three study groups (Sham, 50% PH, 70% PH) during post-surgical phase; animal numbers (Sham/ 50% PH / 70% PH): post-surgery (n = 10/63/59), POD 1 (n = 10/63/59), POD 2 (n = 10/42/39), POD 3 (n = 10/42/38), POD 4 (n = 10/21/18), POD 5 (n = 10/21/18), POD 6 (n = 10/21/18), POD 7 (n = 10/21/18); two-way ANOVA F(14,621) = 4.026, p < 0.0001; POD2: Sham vs. 50% PH padj = 0.0125 and Sham vs. 70% PH padj = 0.0005; POD3: Sham vs. 50% PH padj = 0.0002 and Sham vs. 70% PH padj = < 0.0001; POD4: Sham vs. 70% PH padj = 0.03 and 50% PH vs. 70% PH padj = 0.0313; POD5: Sham vs. 70% PH padj = 0.0072 and 50% PH vs. 70% PH padj = 0.0011; POD6: Sham vs. 70% PH padj = 0.0105 and 50% PH vs. 70% PH padj = 0.0001; POD7: Sham vs. 50% PH padj = 0.0250 and 50% PH vs. 70% PH padj < 0.0001.

Former Figure 5:

Fig 4. Boxplots of severity scores (body weight, general condition, wound healing, and spontaneous locomotor behavior) after Sham or PH with upper limits with a gradual allocation of severity (mild: ≥ 5 points, moderate: ≥ 10 points, and severe: ≥ 20 points); animal numbers (Sham/ 50% PH / 70% PH): POD 1 (n = 10/63/60), POD 2 (n = 10/42/39), POD 3 (n = 10/42/38), POD 4 (n = 10/21/18), POD 5 (n = 10/21/18), POD 6 (n = 10/21/18), POD 7 (n = 10/21/18); Kruskal–Wallis-test (χ2 = 353.3, p < 0.0001, degrees of freedom (df) = 21); POD1: Sham vs. 50% PH and Sham vs. 70% PH padj < 0.0001; POD2: Sham vs. 50% PH and Sham vs. 70% PH padj < 0.0001; POD3: Sham vs. 50% PH padj = 0.0253 and Sham vs. 70% PH padj < 0.0001.

Former Figure 7:

Figure 6: Comparison of fecal corticosterone metabolites (FCMs) in samples weaned during Open Field or during anesthetic induction in various groups. (A) Percentage changes in FCM levels in the three groups over time; sampel sizes (Sham/ 50% PH / 70% PH) on each time point: baseline (n = 10/61/62), surgery (n = 10/54/57), POD 1 (n = 9/54/55), POD 3 (n = 10/40/36), POD 4 (n = 10/20/18), POD 7 (n = 10/19/18); F(10,528) = 2.062, p < 0.0001; surgery: 50% PH vs. 70% PH padj = 0.0031; POD1: Sham vs. 70% PH padj = 0.0377 and 50% PH vs. 70% PH padj = 0.0299; POD3: Sham vs. 50% PH padj = 0.0169; Sham vs. 70% PH padj < 0.0001 and 50% PH vs. 70% PH padj = 0.0273; POD4: Sham vs. 50% PH padj = 0.0137 and Sham vs. 70% PH padj < 0.0001. (B) Standardization of FCMs levels (μg)/g feces in relation to liver weight (calculated by BW-LW-ratio) at POD1; sample sizes: Sham n = 9, 50% PH n = 54, 70% PH n = 55. For liver weight, a value of 4.15% of total body weight was calculated (own data). Data were checked for normality and homoscedasticity using the Shapiro–Wilk and the Levene test. Owing to the observed variance between groups (p = 0.007), the non-parametric Kruskal-Wallis test was used (χ2 = 20.9, df = 2, p < 0.0001). The observed between-group differences were analyzed using the Dunn's post-hoc test and the false-discovery rate for multiple tests was controlled using the Benjamini-Hochberg criterion. Significant differences were found between the 50% and the 70% PH group (Z = -3.727, padj = 0.0006) as well as between the 70% PH and the Sham group (Z = 3.541, padj = 0.0006).

Former Figure 9:

Fig 8. Serum measurements. Boxplots of serum (A) alanine aminotransferase (ALT) and (B) aspartate aminotransferase (AST) levels (U/L) showing group-individual values in group comparisons (Sham, 50% PH, 70% PH); Sham: TI (n = 9), laparotomy (n = 10), eutha POD 7 (n = 10); 50% PH: TI (n = 12), liver resection (n = 63), eutha POD 1 (n = 21), eutha POD 3 (n = 21), eutha POD 7 (n = 21); 70% PH: TI (n = 12), liver resection (n = 61), eutha POD 1 (n = 18), eutha POD 3 (n = 20), eutha POD 7 (n = 18); ALT: one-way ANOVA F(12,277) = 54.71, p < 0.0001; POD1: 50% PH vs. 70% PH padj < 0.0001. AST: one-way ANOVA F(12,276) = 26.05, p < 0.0001; POD1: 50% PH vs. 70% PH padj < 0.0001.

In line 247 the authors mention that statistical analyses have been performed to compare their data with the baselines. However, no results of these analyses are presented in the manuscript. We fully agree that this sentence is superfluous. Of course, we performed the corresponding analysis and evaluations within each group, against its respective baseline, in addition to group comparison. However, many of the parameters are subject to physiological changes that occur in the course of chronical trials. For example, changes in body weight due to growth and gain, and changes in blood count due to liver resection. Since significant changes were determined, which we did expect in one way, but do not provide any added value for the assessment and grading of "moderate" severity grades, these graphs were not shown. Rather, our focus was on examining and reporting gradations and differences within a "moderate" load (represented by the three groups).

In order not to confuse the reader, we have therefore decided to delete the sentence “… Tukey´s and Dunnett´s tests were used for multi-group comparisons (e.g., comparison with Sham group) or comparison with baseline (Dunnett´s). …” (line 247) from the manuscript.

Line 286: I would suggest to write “end of study” instead of “endpoint”. The difference between study end and humane endpoint becomes clearer this way. We are grateful for this hint and changed the wording in the corresponding line 319 of the revised manuscript to:

“… all animals reached the intended end of study; …”

Line 291: The authors describe that 2 animals of the 70% PH group died in the nights between POD1 and POD3 for unknown reasons. The EU working document on a severity assessment framework reads „The actual severity for animals found dead should be reported as 'severe' unless an informed decision can be made that the animal did not experience severe suffering prior to death“. The authors should discuss their decision to categorize PH70% as a manipulation of moderate severity (which is indeed supported by the reviewers) under the aspect of the two animals which were found dead, in particular as the cause of death was unknown. Was data from these animals excluded from the study?

You are absolutely right that the severity in these animals must be classified as "severe", and of course we did this. However, as these animals do not provide a complete set of data and some measurement parameters are falsified (higher severity leading to falsified parameters or values and finally to death), these animals were excluded from the statistical evaluation. This is also evident from the n-numbers given in the legends of the figure (e.g,. new figures 3, 4, or 6).

We have taken your advice into consideration and have added this aspect to the section in the discussion. The new version is as follows:

Line 479 – 481: “… What is evident, however, is that the severity of an animal that dies unexpectedly during its experiments, succumbs to the consequences of its treatments or reaches humane endpoints must always be classified as "severe" unless the opposite can be verified. …”

Line 589 – 590: “… Nevertheless, for the animals that died due to unknown reasons before the end of the study, the severity grade still has to be classified as "severe". …”

Legend of Fig 7B: It should be mentioned that the data shown refer to POD1. We thank you for this hint, improving the manuscripts clarity. We added the time point of POD 1 in the legend of Figure 6B (former figure 7B), accordingly. Please see line 369, of the revised manuscript.

Fig. 8: What is the dashed line in Fig 8B? This should be described in all Figures with PCA containing such a line.

The dashed line in figure 7B (former figure 8B) represents the cut-off for the uniform variance distribution. That means: if they all had the same importance, the parameters would be up to this line. This makes it easier for the reader to interpret whether one variable has more or less "importance" than another, in mean.

We add this information in the legend of the figures in line 387 of the revised manuscript.

Discussion: Line 457f: The authors state that “all three groups exhibited an immediate increase in postoperative severity”. This statement should be re-considered for the following reasons: o No telemetric data are available for the sham group

You are right, the data of SHAM animals are missing. Therefore, we revised the sentence and added the following explanation to the manuscript.

Pease see revised text in line 501-505:

“… All three test groups exhibited an immediate increase in postoperative severity (severity score, body weight change, OF and FCMs) with subtile variations. Further, this was supported by the results of AST, ALT and telemetric data analysis in the PH groups. However, these data are not available for the Sham group to confirm the results. …”

o No ALT and AST values are available for the sham group for the critical test day POD1.

Due to the restriction of our protocol granted by the governmental animal care and use committee in accordance with the 3R principle, the SHAM group we were not allowed to sample and euthanize the animals prior to POD 7. Therefore, no blood, liver weight or organ proliferation/regeneration could be determined at any other POD than POD 7.

o The severity score of the sham group does not seem to be increased on POD1, POD2, and POD3.

You are right. The graph of former figure 5 represents the score values of each group on its respective POD. Upper and lower limits are displayed, as well as the majority distribution of the score points. Thus, the lowest value is always 0 (no severity) and the highest is always a maximum of 20 (human endpoint). Hence, the animals of the SHAM group had a median score of "0" at POD 1-3 and within the group a “standard deviation” of POD 1: 1.5 points, POD 2: 2 points and POD 3: 2 points. However, since the graph always displays the highest measured score value of a group as the "upper limit", and this was 5 points for the SHAM group on all 3 days, the severity level of this group does not seem to change or increase within this time.

Nevertheless, it must be taken into account at this point that the score per se is not sensitive enough to detect the below threshold (very mild severity).

Line 459: The results on severity scoring discussed in this paragraph are hard to interpret as the composition of the score is not described properly before. If the score is influenced heavily by wound healing parameters, one must ask if the resulting scoring really justifies to draw conclusions about mild/moderate/severe severity from that score as found in figure 5. Maybe the authors could discuss if the used severity score in fact presents overall burden on the animal due to general anesthesia and surgery or if the big influence of the category wound healing is leading to a wrong classification in higher severity grades.

You are right that the result here seems difficult to interpret. However, there is no doubt that a liver resection, and thus a visceral surgery performed via laparotomy, is possible without a corresponding abdominal wound. Of course, this always requires post-surgical wound healing. This is therefore indispensable and must be regarded in severity assessment. However, the wound healing is only one parameter out of 4 parameters of the scoring and therefore 25%. Nevertheless, a severe wound healing problem could lead to the humane endpoint. To clarify this in more detail, we gave detailed information about the score in the additional material methods (see supplemental material).

Line 524f: Could the authors explain why they consider the unsupported rearing as additional perspective to stress assessment? They mentioned that it is affected by abdominal surgery meaning that it is decreased post op. Is it therefore not rather a parameter potentially indicating abdominal pain or discomfort? Do the authors know of results in rearing behavior in anesthetized/untreated rats to distinguish between the effects of pain vs. effects of anesthesia or handling? One could ask if the applied analgesic regimen is sufficient for the described laparotomy. Maybe the authors could briefly discuss this point.

We postulate that this parameter is suitable for detecting impairment due to abdominal surgery. It is true that we did not have any anesthesia-only or handling-only animals among the experimental groups to be able to determine an effect in this parameter.

However, a previously published study described that the rearing behaviour is to be interpreted as “normal” influence 2. Here it was reported that the rearing behaviour per se is used as a marker of environmental novelty in unfamiliar surroundings. We therefore base our hypothesis on the assumption that this behaviour, under the influence of the surgery, continues to decrease despite habituation. Apparently, this parameter is negatively influenced by the surgery. Therefore, the counts of rearing behaviour were reduced in favor of remaining on the ground.

Since each animal was already tested for this parameter before laparotomy/liver resection, not only does each animal represent its own baseline, but this can also be seen as an "unaffected" group.

However, to what extent the parameter "unsupported rearing" reacts directly to the liver resection, the laparotomy or even only to the anaesthesia cannot be defined at this point in time. Further studies are needed to evaluate and validate this.

An influence of the parameter by the choice of analgesic and its mode of action is possible, but must also be questioned and checked in subsequent trials.

Reviewer #2:

The use of metamizole as analgesics is good for visceral considerations but does not sound to be sufficient. Why not using opioids in order to improve analgesia? Moreover length of analgesia by metamizole is short since only one administration is performed (or until POD3, not clear, line 190?), which means there is no more pain after 12-24 hours. Why not carrying on 2 to 3 days post-surgery?

We thank the reviewer for this question, as it seems that we have not explained this point clearly enough in the text. We have thoroughly considered the advantages and disadvantages of different analgesia regimes during the experimental design. In the process, we have kept to the recommendations of the GV-Solas for pain therapy in laboratory animals.

The Gesellschaft für Versuchstierkunde / Society of Laboratory Animal Science (GV-SOLAS) is a registered association that is dedicated to the responsible interaction with laboratory animals. It acts as a mediator between animal welfare and research on behalf of humans and animals. In this context, it established a recommendation on pain management in laboratory animals based on German guidelines. 3

So, the reason why we decided against the use of opiates, such as buprenorphine, is due to two main factors. Firstly, we see the sedating effect of opiates as a major influencing factor in the assessment of behavioural experiments. Furthermore, depending on the time of drug administration, these influences sometimes modulate activity in either direction. In addition, opiates often cause undesirable side effects such as pica behaviour in rats.4 This can lead to digestive disorders and even increased wound manipulation (up to wound dehiscence). This would again influence severity assessment and the registered score values.

The use of NSAIDs could also not be implemented due to the experimental investigation of inflammatory parameters.

On the other hand, these side effects do not occur with the use of metamizole.

All animals were therefore injected with Metamizol s.c. prior to surgery. In order to cover the post-surgical need for analgesia and to avoid having to inject the animals several times a day, the medication was administered via sweetened drinking water.

Since the sweetened drinking water was already offered from the time of housing, all animals were familiar with the modified taste, so that water intake was unchanged.

In the afternoon of the third post-surgical day (POD 3), the metamizole-enriched drinking water was replaced with regular sweetened drinking water, so that during the performance of behavioural tests on day 4 (POD 4), all animals were assessed without analgesia.

For clarity, we have therefore added further information in the revised version. Please refer to line 201 – 207 in the revised version:

“Analgesia (metamizole, Novaminsulfon-ratiopharm® 1 g/2 mL; 400 mg/kg/day) was administered subcutaneously prior to each surgery. For post-surgical pain management, metamizole was administered until the late afternoon of the third day after surgery (POD 3), via sweetened drinking water. This should enable an assessment on the morning of the fourth post-surgical day (POD 4), without the influence of analgesic medication.“

Why are only male rats used? What is the explanation?

We apologize that the explanation for the use of male animals was not addressed clearly enough in the text. In a previous systematic review regarding liver resection in the rat model, we were able to point out that male Wistar rats are the most frequently used rat model in this research area. In order to achieve the greatest transferability to the current research area, we have therefore used male animals only in our study.

To state this more clearly in the text, we have already referred to the corresponding reference in line 61 and 65.

See reference 5 of the revised manuscript:

Zieglowski L. Systematic review: Liver resection in rats in animal-based research - does an optimal model exist? M Sc Thesis, RWTH Aachen University 2019 Prospero: CRD42019122598.

However, we are also aware of a gender gap here. Therefore, we performed a further study to investigate the gender effect within this experimental model. Thank you for rising this point.

Is it really useful to use antibiotics for such a surgery? If yes, is it really useful to use a 2nd/3rd generation of antibiotics?

In general, you are right that the precautionary and routine use of antibiotics should be reconsidered and even avoided. As long as sterile work and sterile materials are used, the use of antibiotics is not absolutely necessary.

In our case, however, telemetry transmitters were implanted in some animals, as indicated. These telemetry transmitters were implanted and re-used several times in different animals after thorough cleaning and disinfection. However, as these were not sterilized, we administered an antibiotic treatment covering the animals to prevent wound infections or bacterial transmission. The administered antibiotic is also used in humans for similar indications 5. In order to treat all animals equally in the sense of comparability of the groups, antibiotics were administered to all animals for three postoperative days after surgery.

In the meantime, we have established and published a protocol with which we can re-sterilize already implanted and used single-use transponders via H2O2 gassing 6. We hope that this procedure will make the use of antibiotics no longer necessary.

Line 177: could you precise the exact percentage of liver withdrawn? LLL + CL + ML make 78 % (not 70 %). Same for 50 %: it is not clear.

We would like to point out that we made a mistake in reporting the resected liver lobes in the 70% group, for which we sincerely apologize. We therefore thank you very much for the critical review of the text and the kind advice. For the resection of 70% of the liver weight (marked in purpel), we only resected the left lateral lobe (LLL) and both median lobes (ML) which lead to a total sum of 30%+38% = 68%. For resecting 50% of the liver weight (marked in green), we resected the left lateral lobe (LLL), left part of the medial hepatic lobe (LML), and both caudate lobes (CL), which lead to a total sum of 30%+12%+10% = 52%. (Assuming that the ratio of LML to RML is approximately 1/3 to 2/3).

Liver lobe

Sub lobes

Total liver mass (%) Left lateral lobe (LLL)

- ~30

Median lobes (ML) left median lobe (LML) = ~12

right median lobe (RML) = ~26 ~38

Right liver lobes (RLL)

inferior right lobe (IRL)

superior right lobe (SRL)

~22

Caudate lobes (CL) anterior caudal lobe (ACL) posterior caudal lobe (PCL)

~10

To clarify this to the reader, we deleted the “CL” in line 185 and further added the following supplementary text to line 185 - 189 of the revised manuscript:

“The focus was not to resect the exact amount of 50% or 70% of the liver mass, but rather to find approximate values that at the same time represent a significant difference between the resected groups, which was based on the most frequently used resection models and resected liver lobes, according to Zieglowski et al. 7.”

Furthermore, we revised former figure 3 (now figure 2) according the raised hint:

Fig 2. Short name of individual liver lobes (visceral view); From left to right: native liver (Sham), 50% PH, and 70% PH; resected liver lobes shown in red with resection line in the resection groups.

Line 458: typo (subtle)

We thank you for this advice. Although it is possible to write the word "subtle" as well as "subtile", we have decided to adapt it on the basis of your comment in order not to confuse the reader and to apply a more comprehensible writing style. Please see revised version of the manuscript, line: 503.

Why do you use FCM and not blood corticoide levels? The explanation given in the discussion should be clarified.

We would like to point out that we not only examined FCMs but also blood corticosterone values. This is described in the “Material and Methods” section under " Measurement of FCMs". We agree that this data is not presented in the manuscript. This is due to the fact that we had very high expectations for this parameter in the experimental design, but it turned out to be too unreliable for evaluation. The evaluation of the blood values and data showed that the blood corticosterone levels were too often below the threshold of 6.1 ng/ml, which made it impossible to calculate exact values (even with dilution of the serum). The remaining values that could be measured, however, had such a high variation that a meaningful statistical evaluation was not possible. Therefore, these values were not presented in this manuscript. For the sake of completeness, however, this was indicated in the text, as the quantities of blood taken must be taken into account as possible factors influencing severity grades. However, the raw data are available via the online data repository.

Could you precise the exact interest of telemetry in your experiences?

The main interest of telemetry was to obtain supplementary objective measurement data that could not be obtained without manipulation, and thus without a possible bias of sampling the data. Since we started with this approach only 24 months ago, we could not rely on our own historical values. Therefore, the work presented here is the follow-up study to Zieglowski et al. 2019. When planning the experiments, it was not certain what actual distress or burden the animals would have to face with, and in order to make this as low as possible, but still as evaluable as possible, in the sense of the 3Rs, only 4 of the 7 animals in a 7-day survival group had the telemetry transmitters implanted.

When selecting the transponder model, we chose one that can measure the following parameters: internal body temperature, activity, heart rate and blood pressure. We implanted the transponder in the flank instead of the abdomen in order to avoid falsifying the severity level of the animals by performing a double/second laparotomy.

However, due to the subcutaneous position of the transponder in the flank and the group housing of the animals, the measured values of the internal body temperature could not be determined correctly, as these were changed by the more superficial position lying in contact to each other.

The activity of an animal can also be recorded by the transmitter. Unfortunately, this measurement does not determine the type of movement nor its quality. It only determines counts that are calculated by the distance to the receiver plate. However, as this value is easily affected and biased by group housing, removing for scoring or performing the behavioural test, it was not taken into account in our evaluation.

Thus, only the two parameters heart rate and blood pressure are representative and were evaluated for the analysis and reported in this manuscript.

In long term, it is envisioned to rank the investigated parameters in order to find out which parameters actually have the most expressive power and influence in this set-up. We agree that this is essential to prevent incorrect interpretations. 8.

Whether telemetry will be part of future studies cannot be answered at this stage, but it has been implemented in the context of an all-encompassing and objective data collection.

We hope that our answers meet the reviewers´ expectations and that the revised version is now acceptable for publication in your esteemed journal.

With best regards,

Sincerely,

Rene H. Tolba

Corresponding author

Additional Literature:

1. Jasiecka A, Maślanka T and Jaroszewski JJ. Pharmacological characteristics of metamizole. Polish journal of veterinary sciences. 2014; 17: 207-14.

2. Lever C, Burton S and O'Keefe J. Rearing on hind legs, environmental novelty, and the hippocampal formation. Reviews in the neurosciences. 2006; 17: 111-33.

3. Margarete Arras KB, Alessandra Bergadano, Mattea Durst EE-S, Thea Fleischmann, Jörg Haberstroh PJ, Martin Sager and Claudia Spadavecchia DZ. Fachinformation "Schmerztherapie bei Versuchstieren". Ausschuss für Anästhesie der GV-SOLAS in Zusammenarbeit mit dem Arbeitskreis 4 in der TVT. 07-2020.

4. Clark JA, Jr., Myers PH, Goelz MF, Thigpen JE and Forsythe DB. Pica behavior associated with buprenorphine administration in the rat. Laboratory animal science. 1997; 47: 300-3.

5. AWMF AK-Pd. Leitlinien zur Hygiene in Klinik und Praxis "Perioperative Antibiotikaprophylaxe".

6. Zieglowski L, Kummecke A, Ernst L and Tolba RH. Re-sterilisation of single-use telemetric devices ESR. 2021.

7. Zieglowski L. Systematic review: Liver resection in rats in animal-based research - does an optimal model exist? M Sc Thesis, RWTH Aachen University 2019 Prospero: CRD42019122598.

8. Bruch S, Ernst L, Schulz M, Zieglowski L and Tolba RH. Best variable identification by means of data-mining and cooperative game theory. Journal of biomedical informatics. 2021; 113: 103625.

---

## [Decision Letter · Decision Letter 1]

29 Jun 2021

PONE-D-21-06410R1

Assessing the severity of laparotomy and partial hepatectomy in male rats – a multimodal approach

PLOS ONE

Dear Dr. Zieglowski,

Thank you for submitting your manuscript to PLOS ONE. After careful consideration, we feel that it has merit but does not fully meet PLOS ONE’s publication criteria as it currently stands. Therefore, we invite you to submit a revised version of the manuscript that addresses the points raised during the review process.

As you can see, the reviewers appreciated the changes that you made and only minor corrections are needed at this step.  

We look forward to receiving your revised manuscript.

Kind regards,

Pavel Strnad

Academic Editor

PLOS ONE

Journal Requirements:

Additional Editor Comments (if provided):

Reviewers' comments:

Reviewer's Responses to Questions

**Comments to the Author**

1. If the authors have adequately addressed your comments raised in a previous round of review and you feel that this manuscript is now acceptable for publication, you may indicate that here to bypass the “Comments to the Author” section, enter your conflict of interest statement in the “Confidential to Editor” section, and submit your "Accept" recommendation.

Reviewer #1: (No Response)

Reviewer #2: All comments have been addressed

2. Is the manuscript technically sound, and do the data support the conclusions?

Reviewer #1: Yes

Reviewer #2: Yes

3. Has the statistical analysis been performed appropriately and rigorously? 

Reviewer #1: Yes

Reviewer #2: Yes

4. Have the authors made all data underlying the findings in their manuscript fully available?

Reviewer #1: Yes

Reviewer #2: Yes

5. Is the manuscript presented in an intelligible fashion and written in standard English?

Reviewer #1: Yes

Reviewer #2: Yes

6. Review Comments to the Author

Reviewer #1: The authors fully adressed all of our questions and remarks. We think that their corrections have significantly improved the manuscript.

Some issues were clarified completely, but we have some minor remaining remarks.

1. Thanks for including the severity score sheet. This is very helpful for the reader. We would recommend to refer to the score sheet in the supplementary material in the method section on severity scoring. Otherwise this sheet might be overlooked.

2. Line 209 Additional surgery in animals with wound closure problems: Thanks for describing the procedure of re-closure of surgical wounds. We wonder how you handled the data gathered from the animal that was subjected to an additional anesthesia? Were they included in the results? The authors should state whether they think the second surgery is influencing there results and if this animal should be possibly excluded from the study. Maybe a short sentence could be added on the subject.

3. Blood parameters: the authors gave a satisfying answer to our question why blood parameters were applied. This should be included in the manuscript.

4. Metamizol treatment Line 203ff: Please check the Metamizole dosage in the revised manuscript. Was the s.c. dosage equal to the dosage in the drinking water?

Reviewer #2: Thank you to the authors who have now answered to all my questions. The paper is ready for publication.

7. PLOS authors have the option to publish the peer review history of their article (what does this mean?). If published, this will include your full peer review and any attached files.

Reviewer #1: No

Reviewer #2: **Yes: **Dr. Thomas HUBERT, DVM, PhD, HDR, Dip. LAS, Associate Professor in Surgery

---

## [Author Response · Author response to Decision Letter 1]

9 Jul 2021

Response letter PONE-D-21-06410

Dear Madam/Sir:

First of all, we would like to thank the Editors and the Reviewers again for their valuable input to further improve our manuscript. 

Below, you will find our response to the individual comments of the 1st reviewer as a point-to-point response letter. 

Review Comments to Author:

1. Thanks for including the severity score sheet. This is very helpful for the reader. We would recommend to refer to the score sheet in the supplementary material in the method section on severity scoring. Otherwise this sheet might be overlooked.

Thank you very much for this important recommendation. We added a note, to refer to the supplementary materials, in lines 203 – 205 of the revised manuscript. 

“For severity scoring, a modified version of the scoring system described by Morton et al. (1985) [12] was used daily to assess the general condition of rats (see complete score sheet in supplementary material).”

2. Line 209 Additional surgery in animals with wound closure problems: Thanks for describing the procedure of re-closure of surgical wounds. We wonder how you handled the data gathered from the animal that was subjected to an additional anesthesia? Were they included in the results? The authors should state whether they think the second surgery is influencing there results and if this animal should be possibly excluded from the study. Maybe a short sentence could be added on the subject.

Yes, data of re-sutured animals were included in the analysis. We think, that wound healing disorders and surgery-related complications are a physiological bias to a certain extent, which should not be excluded. Furthermore, these wound healing disorders were pre-defined in the score sheet and do not constitute a reason for the exclusion of animals from both, animal welfare and a research perspective. In addition, biological variability of about 20% was already considered in the design of the experiment. In conclusion, even under the impact of these data (9 out of 136 included animals, ≤ 7%), the severity grades were evaluated as low and less severe than assessed by the EU-Directive. 

Please see lines 212 – 219 of the revised Material and Methods section:

“To assess wound healing, several stages with subsequent proceedings were defined. If only the thread ends were gnawed off (wound edges adapted), a more intensive monitoring was performed. If the adaptation of wound edges was about to loosen, a wound clip was applied on the corresponding area to secure the suture, and was removed as soon as the wound looked resilient. If a suture dehiscence of single knots occured, the animal was re-anesthetized, the wound was cleaned, and re-sutured. In the case that the suture of the abdominal muscle layer ruptured (hernia; skin suture closed), the animal was re-anesthetized, skin sutures were removed and finaly muscles layer and skin were re-sutured.”

Further, in lines 295 - 297:

“Data of re-sutured animals or animals with surgery-related complications were included in the analysis, as long as they did not reach the predefined humane endpoints.”

And lines 325 – 329 of the revised Results:

“Further wound healing disorders occurred in the post-surgical recovery period: two skin sutures had to be fixed with additive wound clips (PH70%); one skin suture had to be re-sutured under short general anesthesia (SHAM group); six animals got a hernia of the muscle layer and had to be re-sutured under general anesthesia (PH70%). Overall, the scores …”

To make this approach transparent to the reader, we added the following text to the discussion. Please see lines 519 – 522:

“Although, data of re-sutured animals were included in the analysis, we think that wound healing disorders and surgery-related complications are a physiological bias to a certain extent, which should not be excluded. Even under the impact of these data (9 out of 136 included animals, ≤ 7%), the severity grades were evaluated as mild to moderate.”

3. Blood parameters: the authors gave a satisfying answer to our question why blood parameters were applied. This should be included in the manuscript.

We thank you for this improving hint and hope, that the additional information points out why blood parameters were assessed at all. Therefore, we included the sentences in the results and the discussion section of the manuscript.

Please see lines 407 - 409 of the revised manuscript:

“… Metamizole is known to cause agranulocytosis in patients [18], which we could not confirm in our animals. To identify and verify metamizole-dependent changes, which could have an impact on severity grades, the blood cell counts were assessed. …”

And line 505 – 507: 

“While surgical interventions can lead to inflammation and thus to changes in the white blood cell counts, bleeding that may be caused by liver resection, can lead to changes in the red blood cell counts.”

4. Metamizol treatment Line 203ff: Please check the Metamizole dosage in the revised manuscript. Was the s.c. dosage equal to the dosage in the drinking water?

We apologize for the misleading wording. We added detailed information to each application time point to clarify the different treatments and dosages. Please see lines 191 – 196:

“Analgesia was administered subcutaneously (metamizole, Novaminsulfon-ratiopharm® 1 g/2 mL; 100 mg/kg, single dose) prior to each surgery. For post-surgical pain management, metamizole (Novaminsulfon-ratiopharm® 1 g/2 mL; 400 mg/kg/day, oral) was administered until the late afternoon of the third day after surgery (POD 3), via sweetened drinking water.”

We hope that our answers meet the reviewers´ expectations and that the revised version is now acceptable for publication in your esteemed journal.

With best regards,

Sincerely,

Rene H. Tolba

Corresponding author

---

## [Editor Report · Decision Letter 2]

12 Jul 2021

Assessing the severity of laparotomy and partial hepatectomy in male rats – a multimodal approach

PONE-D-21-06410R2

Dear Dr. Zieglowski,

We’re pleased to inform you that your manuscript has been judged scientifically suitable for publication and will be formally accepted for publication once it meets all outstanding technical requirements.

Kind regards,

Pavel Strnad

Academic Editor

PLOS ONE
---

## [Editor Report · Acceptance letter]

21 Jul 2021

PONE-D-21-06410R2 

Assessing the severity of laparotomy and partial hepatectomy in male rats – a multimodal approach 

Dear Dr. Zieglowski:

I'm pleased to inform you that your manuscript has been deemed suitable for publication in PLOS ONE. Congratulations! Your manuscript is now with our production department. 

Kind regards, 

on behalf of

Dr. Pavel Strnad 

Academic Editor

PLOS ONE